

Effects of Liquid Phase Cloud Microphysical Processes in Mixed
Phase Cumulus Clouds over the Tibetan Plateau
Xiaoqi Xu[1], Chunsong Lu[1*], Yangang Liu[2], Wenhua Gao[3], Yuan Wang[1], Yueming
Cheng[1], Shi Luo[1], Kwinten Van Weverberg[4]
1. Key Laboratory for Aerosol-Cloud-Precipitation of China Meteorological
Administration/Collaborative Innovation Center on Forecast and Evaluation of
Meteorological Disasters (CIC-FEMD), Nanjing University of Information Science &
Technology, Nanjing, China
2. Environmental and Climate Sciences Department, Brookhaven National Laboratory,
Upton, US
3. State Key Laboratory of Severe Weather, Chinese Academy of Meteorological
Sciences, Beijing, China
4. Met Office, Exeter, UK
*    Correspondence: luchunsong110@gmail.com



16        Abstract

17        Overprediction of precipitation over the Tibetan Plateau is often found in

numerical simulations, which is thought to be related to coarse grid sizes or inaccurate
large-scale forcing. In addition to confirming the important role of model grid sizes,
this study shows that liquid-phase precipitation parameterization is another key culprit,
and underlying physical mechanisms are revealed.

22        A typical summer plateau precipitation event is simulated with the Weather

Research and Forecasting (WRF) model by introducing different parameterizations of
liquid-phase microphysical processes into the commonly used Morrison scheme,
including autoconversion, accretion, and entrainment-mixing mechanisms. All
simulations can reproduce the general spatial distribution and temporal variation of
precipitation. The precipitation in the high-resolution domain is less overpredicted than
in the low-resolution domain. The accretion process plays more important roles than
other liquid-phase processes in simulating precipitation. Employing the accretion
parameterization considering raindrop size makes the total surface precipitation closest
to the observation which is supported by the Heidke skill scores. The physical reason
is that this accretion parameterization can suppress fake accretion and liquid-phase
precipitation when cloud droplets are too small to initiate precipitation.



## 1. Introduction

The Tibetan Plateau (TP) is the highest and largest plateau of the world with an average elevation of more than 4 km above the sea level and an area larger than $2.5 \times 10^6$ km$^2$. Its active exchanges of heat and moisture have significant influences on climate and environmental change, not only in China but also over East Asia and even the entire northern hemisphere through strong thermal and dynamic forcing (Yeh, 1950; Flohn, 1957; Hahn and Manabe, 1975; Ye, 1981; Wu and Chen, 1985; Yanai et al., 1992; Ding et al., 2001; Wang et al., 2008; Molnar et al., 2010; Yang et al., 2014). Many studies have reported that the tropospheric heating over the TP has decisive effects on the maintenance of the Asian summer monsoon (Luo and Yanai, 1983, 1984; Ueda and Yasunari, 1998). The upward transport of sensible heat and the release of latent heat over the plateau region due to convective clouds are important heat sources in the upper troposphere, driving the East Asian summer monsoon and associated precipitation (Nitta, 1983; Luo and Yanai, 1984; Yanai and Li, 1994; Ueda et al., 2003; Hsu and Liu, 2003).

During the summer monsoon, deep convection develops over the TP with a marked diurnal cycle in precipitation (Fujinami and Yasunari, 2001; Kurosaki and Kimura, 2002; Chen et al., 2017a), frequently associated with mesoscale vortices (Shen et al., 1986; Wang et al., 1993; Li et al., 2008). Overall, summer precipitation on the plateau is characterized by frequent, but rather weak convection (Gao et al., 2016). Undoubtedly, these characteristics are heavily influenced by the unique terrain of the



TP (Porcù et al., 2014; Chen et al., 2017b;Wu and Liu, 2017).
The particularity of the TP has led it to be one of the most challenging areas for
precipitation simulation. Precipitation simulated with coarse resolution (>3km) is often
found to be higher than observations (Maussion et al., 2011; Xu et al., 2012; Gao et al.,
2016). Some studies claimed that low resolution was responsible for the overprediction
of precipitation (Sato et al., 2008; Xu et al., 2012). Sato et al. (2008) also showed that
a finer resolution simulation was more efficient in reproducing the diurnal variation of
summer precipitation. Maussion et al. (2011) investigated effects of different physical
schemes and found a strong microphysical sensitivity for convective precipitation, but
much smaller sensitivity for simulations with dominant advection over the TP. Gerken
et al. (2013) compared simulations with different forcing data and found that there were
large differences in the precipitation generated from different initial and boundary
conditions. Some studies also claimed that elevated aerosol concentrations can
remarkably enhance convections due to specific topography of the TP, however, few
studies focus on this issue (Zhou et al., 2017) which was broadly investigated in other
areas (e.g. Wang et al., 2011; Fan et al., 2018).
The high elevation of the TP, and hence the typically low melting level, enables
plenty of supercooled liquid water, even in summer (Gao et al., 2016; Zhao et al., 2017;
Tang et al., 2019). Hence, it is likely that liquid precipitation processes play a role in
the precipitation overestimation in this region. For instance, Zhao et al. (2017)
confirmed that supercooled cloud water dominated in precipitating cumulus clouds over



the Naqu area at the temperature of -2.5 to -3.5°C. By analyzing the raindrop size
distribution at Maqu over the TP , Li et al. (2006) argued that the liquid-phase processes
were important for surface precipitation, although ice-phase rain processes dominated
over the region. Gao et al. (2016) investigated the roles of liquid-phase rain
microphysical processes and suggested that liquid-phase rain processes could be
important over the precipitation centers during weak convection over the TP.

Three parameterized liquid-phase processes are investigated in this paper:

autoconversion, accretion, and entrainment-mixing. Autoconversion is expressed as the
mass conversion rate from cloud to rain due to the collision-coalescence of cloud
droplets while accretion is defined as the rate of mass conversion from cloud to rain
due to the collection of cloud droplets by raindrops. The sum of autoconversion and
accretion is calculated as the total mass conversion from cloud to raindrop populations
during the collision-coalescence process (Wood, 2005a). Wang et al. (2012) and
Gettelman et al. (2013) highlighted that autoconversion was important for the initiation
of precipitation whereas accretion was responsible for the amount of precipitation. The
process of entrainment and mixing between cloud and environment is one of the most
uncertain processes in cloud physics. The key issue of the entrainment-mixing process
is whether evaporation due to mixing causes a reduction of only droplet size
(homogeneous mixing), only droplet number (extremely inhomogeneous mixing), or
both. Therefore, different entrainment-mixing mechanisms can affect cloud
microphysical properties and hence cloud-related processes such as radiation and



precipitation (Lasher-Trapp et al., 2005; Grabowski, 2006; Chosson et al., 2007;
Slawinska et al., 2008; Lu et al., 2013;Cooper et al., 2013).

So far, it is still unknown how the above three liquid-phase processes affect

precipitation over the TP and whether improving the parameterizations of these three
liquid-phase processes can mitigate the problem of overpredicted precipitation. Further
unknown is the relative contributions of these three processes to surface precipitation
over the TP and which of these parameterized processes exhibits the largest sensitivity
in terms of surface precipitation. This study fills these gaps by comparing simulations
of a precipitation event over the TP with different liquid-phase parameterizations and
dissecting the underlying physical mechanisms.

This paper is organized as follows: A brief introduction on the precipitation event

and experimental setup are given in section 2. Section 3 discusses the influence of
liquid-phase processes on cloud microphysics, radiation, and precipitation in different
numerical experiments. Summary and conclusions are given in section 4.

**2.  Description of precipitation event and observational dataset**
**2.1 Case description and observations**

As mentioned in Gao et al. (2016), the entire plateau experienced a large frontal

system from 21 to 23 July 2014 and observed precipitation initiated at 0400 UTC
(Coordinated Universal Time) 22 July. The simulations are compared against the data
derived from multiple satellite precipitation data sets and blended using a dynamic



Bayesian model averaging (BMA) algorithm in regions with sparse gauge observations,
proposed by Ma et al. (2018). This new precipitation dataset is more viable for complex
terrains such as the TP region. Hence, observations should be more accurate and have
higher spatial (0.1°) and temporal (1h) resolution than the Tropical Precipitation
Measuring Mission (TRMM) usually used in this region (Fu et al., 2007; Yin et al.,
2008; Maussion et al., 2011; Xu et al., 2012).

**2.2 Model and experiment description**

The Weather Research and Forecasting (WRF) model version 3.8.1 is used to

simulate this typical summer TP precipitation event. The WRF model is a next-
generation mesoscale numerical weather prediction system designed for both
atmospheric research and operational forecasting applications. Here, WRF is used as a
cloud-resolving model with 1 km horizontal grid spacing for the innermost domain
(referred to as domain 03) with 276×276×45 grid points, which covers most of the
plateau center; the spatial resolutions for the two outer domains (01 and 02) are 25 km
and 5 km with 200×200×45 and 176×176×45 grid points, respectively (Figure 1).
Initial and boundary conditions are provided by the National Centers for Environmental
Prediction Final operational global analysis data with 1° spatial and 6 h temporal
resolution. The simulation starts at 1200 UTC 21 July and ends at 0000 UTC 24 July,
with a total of 60 h integration time. We focus on the results of the last 48 h from domain
02 and domain 03 with a 30-minute interval.



The microphysics scheme used in the control run is the Morrison double-moment
scheme Morrison and Grabowski, 2008. Note that this bulk scheme is different from
the default version released in the WRF model with a fixed cloud droplet number
concentration ($N_c$) (e.g. $N_c = 250$ cm$^{-3}$). This version can predict the number
concentration and mass mixing ratios of cloud droplets ($N_c$, $q_c$), raindrops ($N_r$, $q_r$), ice
crystals ($N_i$, $q_i$), snow particles ($N_s$, $q_s$), and graupel particles ($N_g$, $q_g$). The main liquid-
phase conversion processes, i.e. autoconversion ($R_{auto}$; kg m$^{-3}$s$^{-1}$) and accretion ($R_{accr}$;
kg m$^{-3}$s$^{-1}$), are both based on Khairoutdinov and Kogan (2000), further referred to as
the KK schemes:
$$R_{auto} = 1350 \times q_c^{2.47}(N_c \times 10^{-6})^{-1.79}\rho_a^{-1.47}, \tag{1}$$

$$R_{accr} = 67 \times (q_c q_r)^{1.15}\rho_a^{-2.3}, \tag{2}$$

where $\rho_a$ is the air density.
To explore the influences of liquid-phase cloud microphysical processes in mixed-
phase clouds, we implement several different expressions for autoconversion, accretion,
and entrainment-mixing process into the Morrison scheme, and examine the model
sensitivity. In addition to the default KK schemes, three commonly-used
autoconversion schemes are employed and referred to as Be68, Bh94, and LD04 for
convenience, respectively:
1) Berry (1968):
$$R_{auto} = \frac{3.5 \times 10^{-2} q_c^2}{0.12 + 1.0 \times 10^{-12}\frac{N_c}{q_c}}, \tag{3}$$

This is the default scheme in several global climate models such as Model for





Interdisciplinary Research on Climate version 5 (MIROC5; Michibata and Takemura,
2015; Jing and Suzuki, 2018)
2) Beheng (1994):
$$R_{\mathrm{auto}} = 6.0 \times 10^{28} n^{-1.7} (q_c \times 10^{-3})^{4.7} (N_c \times 10^{-6})^{-3.3}, \qquad (4)$$
where $n$ is set to 10 in Eq.4, which is related to the width of cloud droplet size
distribution;
3) Liu and Daum (2004):
$$R_{\mathrm{auto}} = P_0 T \, , \qquad (5a)$$
$$P_0 = 1.1 \times 10^{13} \left[ \frac{(1+3\varepsilon^2)(1+4\varepsilon^2)(1+5\varepsilon^2)}{(1+\varepsilon^2)(1+2\varepsilon^2)} \frac{q_c^3}{N_c} \right], \qquad (5b)$$
$$T = \frac{1}{2}(x_c^2 + 2x_c + 2)(1 + x_c)e^{-2x_c}, \qquad (5c)$$
$$x_c = 9.7 \times 10^{-14} N_c^{3/2} q_c^{-2}. \qquad (5d)$$
The LD04 derived by Liu and Daum (2004) and Liu (2005) considers relative
dispersion $\varepsilon$ (the ratio of the standard deviation to the mean radius) in addition to
droplet concentration and liquid water mixing ratio. This scheme was implemented into
the WRF double-moment schemes (Xie and Liu, 2011; Xie et al., 2013). $P_0$ and $T$
represent rate function and threshold function, respectively; the $\varepsilon$ is set to 0.4 as the
average value based on Zhao et al. (2006) and Wang et al. (2019).

Considering that most accretion schemes only take mass mixing ratios of cloud

droplets and raindrops (i.e. $q_c$ and $q_r$) into account, a parameterization that relates the
accretion process to liquid droplets number concentration and drop size distribution is
adopted from Cohard and Pinty (2000), named as CP2k:



$$R_{\mathrm{accr}} = \frac{\pi}{6}\rho_{\mathrm{W}}\rho_{\mathrm{a}}K_1 \frac{N_{\mathrm{c}}N_{\mathrm{r}}}{\lambda_{\mathrm{c}}^3}\left(\frac{A_1}{\lambda_{\mathrm{c}}^3} + \frac{B_1}{\lambda_{\mathrm{r}}^3}\right),$$

$\quad if\ R_{\mathrm{r}} \geq 50\ \mu m,$ and $\qquad\qquad (6a)$
$$R_{\mathrm{accr}} = \frac{\pi}{6}\rho_{\mathrm{W}}\rho_{\mathrm{a}}K_2 \frac{N_{\mathrm{c}}N_{\mathrm{r}}}{\lambda_{\mathrm{c}}^3}\left(\frac{A_2}{\lambda_{\mathrm{c}}^6} + \frac{B_2}{\lambda_{\mathrm{r}}^6}\right),$$

$\quad if\ R_{\mathrm{r}} < 50\ \mu m,$ $\qquad\qquad (6b)$
where $\rho_{\mathrm{W}}$ is the water density, $R_{\mathrm{r}}$ is the raindrop radius, $K_1$ and $K_2$ are empirical
constants; the subscripts $c$ and $r$ denote cloud droplets and raindrops, respectively. $A_1$,
$A_2$, $B_1$, and $B_2$ are the functions related to two dispersion parameters of the gamma size
distribution; $\lambda$ is the slope parameter and is derived from the dispersion parameter,
number concentration and mixing ratio of the species (see Morrison et al., 2005). Due
to specified dispersion parameters for raindrops, $\lambda_{\mathrm{r}} = (\pi\rho_{\mathrm{W}}N_{\mathrm{r}}/q_{\mathrm{r}})^{1/3}$ which is
inversely proportional to the radius of the raindrops. Another accretion scheme (Ko13,
Kogan, 2013) is also tested:
$$R_{\mathrm{accr}} = 8.53 \times q_c^{1.05}q_r^{0.98}\rho_a^{-2.03},\qquad\qquad (7)$$
For the entrainment-mixing process, the subgrid-scale mixing can be defined using
a single parameter α in this microphysical scheme (Morrison and Grabowski, 2008; Lu
et al., 2013):
$$N_c = N_{\mathrm{c0}}\left(\frac{q_{\mathrm{c}}}{q_{\mathrm{c0}}}\right)^{\alpha},\qquad\qquad (8)$$
where the $N_{\mathrm{c}}$ and $N_{\mathrm{c0}}$ are the number concentrations of cloud water droplets after and
before the evaporation process, respectively, and the $q_{\mathrm{c}}$ and $q_{\mathrm{c0}}$ represent the
corresponding mixing ratios, respectively. The parameter α can set to be any value
between 0 and 1 corresponding to a different degree of the subgrid-scale mixing



homogeneity. When α = 0, homogeneous mixing is assumed (the control run). On the
contrary, when α = 1, extremely inhomogeneous mixing is assumed (the INHOMO run).

In total, we have 7 simulations: the control run with the KK schemes for

autoconversion and accretion, and homogeneous mixing mechanism, and sensitivity
tests with three autoconversion schemes (Be68, Bh94 and LD04), two accretion
schemes (CP2k and Ko13), and one entrainment-mixing scheme (INHOMO).

**3. Results**
**3.1  Control run**
**3.1.1 Precipitation from the control run and observations**

Result of 48 h accumulated precipitation over domain 02 (Figure2b) rather than

domain 03 (Figure 2d) is used to compare with observations (Figures 2a and c) because
the domain resolution of 5 km is closer to that of the observation data (0.1°). The
precipitation from 0000 UTC 22 July to 0000 UTC 24 July 2014 from the control run
is averaged to fit the resolution of 0.1°. The results indicate that the control run can
reproduce the primary rainband oriented in the northeast-southwest direction. The
precipitation in most regions is less than 50 mm and the maximum value is
approximately 80 mm in the observation. Although the control run is spatially
consistent with the observation, the maximum precipitation in simulation is about 200
mm, over twice of the observation. Similar biases were reported in Xu et al. (2012) and
Gao et al. (2016); these inconsistencies could be related to the large-scale dynamic





forcing or the model resolution. For domain 03, when the simulated precipitation is
averaged to 0.1°, there are only about 27*27 data points; the data quantity may not be
big enough to compare the spatial distribution of precipitation between simulations and
observations. This could explain the spatial precipitation bias shown in Figures 2c and
2d. Besides spatial comparison, Figures 3a and 3b show the temporal evolutions of
area-averaged hourly precipitation rate from the observation and the control run over
domain 02 and domain 03, respectively. The black solid lines denote the observation
data. The simulations of both domains correlate well with observations in trends, but
the domain 03 is clearly closer to the observations in terms of precipitation rate. Similar
to previous studies (e.g. Xu et al., 2012), the precipitation of domain 02 with a low
resolution is overestimated compared to the observations. The observations for domain
03 show that there are two peaks of precipitation in the local afternoon (UTC + 6 h).
The first precipitation event starts from 0400 UTC 22 and ends at 1800 UTC with the
maximum precipitation rate of 1.0 mm/h attained at 0900 UTC. The other precipitation
peak is weak with the maximum precipitation rate of only about 0.4 mm/h. The control
run shows a slightly smaller precipitation rate than the observation for the first peak
and a slightly larger rate for the second peak. The time of the peaks in the simulations
is about 2 hours later than that in the observations, which was also reported in Gao et
al. (2018). Generally speaking, the control run captures the main features of the
precipitation evolution (the peaks and the trend) but also produces an artificial weak
peak (~0.2 mm/h) at about 0000 UTC 23 which is not observed.




### 3.1.2 Microphysical processes in the control run

Based on the precipitation mentioned above, the microphysics is examined for
different resolutions/periods. For domain 02, considering that the altitude of the
southeastern corner is lower than the other regions, liquid-phase precipitation is
expected to be stronger. Therefore, domain 02 is divided into two parts: the southeastern
corner and the other regions. For domain 03, the two precipitation peak periods are
studied separately.
Figure 4 shows the mean vertical profiles of five types of hydrometeors and their
primary microphysical processes for the two separate regions over domain 02 and the
two precipitation peaks (5 hours) over domain 03, respectively. For domain 02, mixing
ratios of ice-phase hydrometeors (ice, snow, and graupel) and rates of microphysical
processes (RIM-s, RIM-g, MELT) over the southeastern corner (Figures 4c and d) are
generally equivalent to or smaller than those over the other regions (Figures 4a and b).
Mixing ratios of liquid-phase hydrometeors (cloud and rain) and microphysical
processes (ACCR-r, AUTO-r) are larger over the southeastern corner than those over
the other regions. As mentioned above, liquid droplets have more opportunities to grow
over the southeastern corner because of its lower terrain. For domain 03, mixing ratios
of ice-phase hydrometeors (ice, snow, and graupel) and rates of microphysical
processes (EVAP-r, ACCR-s, RIM-s, RIM-g, MELT) are smaller during the second
peak period   (Figures 4g and h) than those during the first peak period (Figures 4e and





f). However, accretion rate of cloud droplets by rain (ACCR-r) is larger for the second
peak than for the first one, although melting is still dominant. Therefore, the liquid-
phase processes over the southeastern corner in domain 02 and the second precipitation
peak in domain 03 are more important than those over the other regions in domain 02
and the first precipitation peak in domain 03, respectively, though the reasons are
different. While ice phase processes are equally important across the entire domain 02,
the warmer temperatures in the lower southeastern corner allow for more liquid phase
precipitation. In domain 03, however, the second peak is clearly associated with smaller
ice-related conversion rates.

**3.2  Sensitivity tests with different parameterizations of liquid-phase processes**

Besides the control run, precipitation, microphysical properties, and their related

processes from the sensitivity simulations are discussed in this section, including Be68,
Bh94, LD04, CP2k, Ko13, and INHOMO.

**3.2.1 Precipitation from the sensitivity tests and observations**

The results of precipitation from the sensitivity tests are shown in Figures 5 and 6.

All simulation cases have produced the similar rain band/trend and precipitation rate,
compared to the control run, except the CP2k experiment. The CP2k has distinctly
weaker precipitation than the other simulations especially over the southeastern corner
in domain 02 and during the second precipitation peak period in domain 03.





Qualitatively, the results from the CP2k are closer to the observations (Figures 2, 3, 5
and 6).

The Heidke skill score (HSS) is used to quantitatively evaluate the simulations

with different schemes:
$$\text{HHS} = \frac{2(ad - bc)}{(a + c)(c + d) + (a + b)(b + d)},\qquad(9)$$
where the four elements *a-d* for HSS, representing the numbers of "hits", "false alarms",
"misses" and "correct negatives", respectively, are calculated from a contingency table
(Table 1). HHS can not only judge well-simulated events (both hits and correct
negatives, element *a* and *d*) but also account for erroneous forecast (*b* and *c*) (Barnston,
1992). A higher HSS (0 ~ 1) represents better skill. As shown in Table 1, $p_t$ is the
threshold value and is set to be 2 mm covering most of the observed and simulated
precipitation area, $p_s$ and $p_o$ are the values from simulations and observations,
respectively.

The elements a-d and HSS for all sensitivity tests over domain 02 and 03 are shown

in Table 2. All the cases in domain 02 have the HSS scores exceeding 0.4 and are close
to each other except for the CP2k. The impacts of changing autoconversion schemes
and mixing mechanisms on HSS are limited. The CP2k accretion scheme, however, has
significantly higher HSS than other cases, particularly due to its high value of d, the
"correct negatives" mainly over the southeastern region for domain 02. The high HSS
scores in the CP2k indicate that changing the accretion scheme is a possible way to
improve the much-overestimated precipitation in simulations over this region. The HSS





scores of all simulations for domain 03 are small because there are too few data points
for evaluation, as mentioned above; slight changes in any of the four factors can cause
a large difference in the final scores. However, the CP2k case still has the highest HSS
of 0.152, much larger than the maximum and mean HSS of other cases, 0.110 and 0.076,
respectively.

**3.2.2 Influences of liquid-phase processes on cloud microphysics**

Table 3 summarizes the microphysical and radiative properties for all the

simulations, including $N_c$, liquid cloud water path (LCWP), cloud optical depth ($\tau$) and
liquid cloud mean effective radius ($\overline{r}_e$) over domain 02 and 03, respectively. Note that
only the cloud data in the grid boxes with hydrometeor mixing ratios larger than 0.01
g/kg are included. The equation for $\tau$ is:
$$\tau = \frac{3}{2}\frac{1}{\rho_w}\int_0^H \frac{\rho_a q_c(z)}{r_e(z)}\,dz,\tag{10}$$

where $q_c(z)$ and $r_e(z)$ are mixing ratio and effective radius of cloud droplets at each
height ($z$), respectively; the extinction efficiency is assumed to equal to 2 (appropriate
at visible wavelengths) (Grabowski, 2006); $H$ is the cloud top height. Because LCWP=
$\int_0^H \rho_a q(z)\,dz$, the column mean of effective radius is given by
$$\overline{r}_e = \frac{3}{2}\frac{LCWP}{\rho_w \tau},\tag{11}$$

All sensitivity tests have effects on cloud microphysics in different ways.

Changing liquid-phase rain formation processes (i.e. parameterizations of
autoconversion and accretion) influences $q_c$ due to their direct effects on the conversion



rates from cloud droplets to raindrops. On the contrary, dilution caused by the
entrainment reduces $q_c$, and the different mixing mechanisms in the subsequent mixing
and evaporation processes determine how many cloud droplets are completely
evaporated.

**3.2.2.1 Autoconversion**

Compared with the control run, the largest differences in all autoconversion cases

are 28.2% (28.0%) in LCWP, 18.1% (18.5%) in $\tau$ and 4.2% (4.78%) in $\overline{r_e}$ over domain
02 (03) mainly due to one order of magnitude difference of autoconversion rate among
different cases (Figures 7a and c). It should be noted that this magnitude of difference
is much smaller than that in typical marine boundary layer clouds, which may have over
three orders of magnitude difference (Wood, 2005a). Considering that the
autoconversion process is indeed sensitive to $q_c$, there are two reasons responsible for
this phenomenon. On the one hand, the temperature of the cloud base over TP region is
low; thus the liquid-phase part of the cloud is thin and cloud droplets do not have
enough vertical distance to grow; on the other hand, the active ice-phase particles can
consume cloud droplets suspended in the supercooled region. Autoconversion is the
initial process to produce raindrops, and thus larger autoconversion rate usually brings
out larger accretion rate (Figures 7b and d).




### 3.2.2.2 Accretion


It is noteworthy that the CP2k scheme has larger differences from the control run
than the three autoconversion cases and the Ko13 case, especially for the LCWP-related
processes. Compared to the control run, differences of the CP2k case over domain 02
(03) are +64.6% (+51.0%) in LCWP, +36.6% (+28.1%) in $\tau$ and +7.9% (+5.6%) in $\bar{r}_e$
while the Ko13 case is much closer to the control run. These large differences are
caused by different accretion intensities in different parameterizations. The CP2k case
has the weakest accretion intensity compared to the other cases. It should be noted that
the weaker accretion process in the CP2k leads to a larger autoconversion rate than that
in the control run, different from the argument mentioned above that stronger
autoconversion leads to stronger accretion. The larger difference between the CP2k and
the control run in domain 02 than in domain 03 is due to the stronger liquid-phase
processes in the southeast corner. Details are discussed in the next section.

### 3.2.2.3 Entrainment-mixing mechanisms


For the entrainment-mixing processes, $N_c$ in the INHOMO run is about 2.6 (4.9)
/cm$^3$ less than the control run, results in (0.9%) 2.4% larger $\bar{r}_e$ over domain 02 (03).
The influence of entrainment-mixing processes on $\bar{r}_e$ is larger than the Be68, the LD04,
and the Ko13, but smaller than the Bh94 and the CP2k. Different from other sensitivity
tests, the influences of entrainment-mixing processes over domain 03 with a higher
resolution are more important than domain 02, since the relevant scales involved in this



process are usually small. The differences between the INHOMO and the control run
are similar to the previous studies using the double-moment microphysics scheme
(Grabowski and Morrison, 2011; Slawinska et al., 2012). As explained in these studies,
entrained air close to saturation is a plausible reason for these small changes (Hoffmann
and Feingold, 2019). It is worth noting that our simulations are concerned with a large
frontal system with a large cloud cover. The relative humidity of grid boxes
experiencing evaporation are mainly larger than 95%.

**3.3   Reasons for the precipitation reduction in the CP2k**
As mentioned before, compared to other experiments, the CP2k exhibits the largest
difference from the control run both for surface precipitation and cloud microphysics.
The reasons are discussed in this section.

**3.3.1 Detailed microphysical processes in the CP2k**
The CP2k experiences an accretion rate that is one to two orders of magnitude
smaller than those in the control run and other simulations (Figures 7b and d). The
weaker accretion process implies that more liquid cloud water remains suspended in
the air and could take part in other microphysical processes such as autoconversion and
riming. As shown in Figures 7a and c, the autoconversion rate in the CP2k is much
larger than that in the control run; the difference is close to the value that applying
different autoconversion schemes directly can cause. Combining two dominant liquid-



phase rain formation processes (autoconversion and accretion), less cloud water is
depleted in the CP2k; as a result, the mean value of LCWP is over 50.0% larger than
that of the control run, as shown in Table 3. Figure 8 shows the vertical profiles of the
mean differences of the dominant conversion process rates between the CP2k and the
control run (CP2k-Control) over the two regions in domain 02 and during the two
precipitation peak periods in domain 03. Similar to Figure 7, the CP2k has a much
smaller accretion rate and larger autoconversion rate. Despite the larger autoconversion
rate, many cloud droplets are suspended above the 0 ℃ isotherm, beneficial for riming
of cloud droplets onto snow or graupel particles (RIM-s + RIM-g). Due to the larger
riming rate, more ice-phase particles melt to more raindrops below the 0 ℃ isotherm
(MELT). Note that the smaller melting rate near 6 ~ 6.5 km in the CP2k over domain
03 is because of the lower melting level in CP2k than in the control run. Table 3 shows
that $\tau$ in the CP2k is larger, which means more solar radiation is reflected to the upper
atmosphere and less short-wave radiation reaches the ground (219.6 W/m$^2$ in the CP2k
vs 226.5 W/m$^2$ in the control run). Such a difference in radiation results in a lower
temperature in the CP2k in the low atmosphere than in the control run. Therefore, the
melting level is lower in the CP2k.

The source of surface precipitation includes both the liquid-phase (mainly ACCR-

r) and the ice-phase (MELT). During the first precipitation peak period in domain 03,
despite of the smaller accretion rate in the CP2k than that in the control run, more riming
leads to more melting. The combination of weaker accretion and more melting in the



CP2k offset each other, and hence the precipitation from the CP2k and the control run
is very close in this period (Figure 6b). A similar chain of events also occurs in domain
02 except for the southeastern corner (Figures 2b and 5d). However, in the control run,
due to relatively low concentration of ice particles during the second peak period in
domain 03, the liquid-phase processes, in particular accretion, become relatively more
important (Figure 4h); for the southeastern corner of domain 02, the large mixing ratio
of cloud droplets even causes the accretion rate to exceed the melting rate (Figure 4d).
Surface precipitation is overestimated in the control run compared with the observations,
as discussed in Section 3.2.1. In the CP2k, the accretion is suppressed which appears to
alleviate the overestimation of precipitation. Therefore, the total surface precipitation
in the CP2k is smaller than that in the control run over the southeastern corner in domain
02 and during the second peak period in domain 03, which is closer to observations.

**3.3.2 Detailed analysis of the CP2k parameterization**
The large differences in cloud microphysics and precipitation between the CP2k
and other cases can be explained based on the different equations for autoconversion
and accretion (Eq. 2, 6 and 7). The different equations for the autoconversion and
accretion can be separated into two basic methods as mentioned in Wood (2005a): the
first one integrates the stochastic collection equation for a wide range of drop size
distributions and then uses a simple power-law fit, such as the KK scheme in the control
run. The second method simplifies the collection kernel and parameterizes the

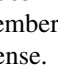
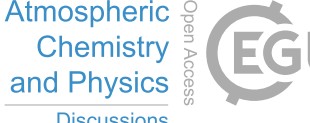

autoconversion and accretion processes, such as the parametrization of the
autoconversion rate in LD04 and accretion rate in the CP2k. Autoconversion schemes
commonly use one of these basic methods. However, the accretion schemes used in
most of the microphysical schemes are based on the first method, and previous studies
largely compare these accretion schemes (Wood, 2005a; Hill et al., 2015). As shown
above and also below, the CP2k accretion rate parameterization is unique and appears
superior to other parameterizations, but this parameterization is only used in a few
microphysics schemes (e.g. WDM6 scheme in WRF, Lim and Hong, 2010).
Figure 9 compares the accretion rate calculated as a function of raindrop radius for
all the accretion schemes under the conditions of $q_c = 1$ g/kg, $R_c = 10$ μm, $N_r = 4000$ /m$^3$.
It is obvious that the three schemes result in different relationships for the accretion rate.
Considering the power-law form in the formula from the first method, i.e., the KK
scheme in the control run and the Ko13 scheme, accretion rate is linearly related to
raindrop radius in the logarithmic space. However, the CP2k accretion rate has an
inflection point at 50 μm due to the piecewise function in Eq. 6. Under the condition of
adequate cloud water, the accretion process in the KK or the Ko13 scheme only depends
on rain water mixing ratio. However, in the CP2k, if the raindrop radius is less than 50
μm, the accretion rate is very small. As shown in Figure 9, the accretion rate in the KK
or the Ko13 scheme is always larger than in the CP2k when the raindrop radius is
smaller than 2000 μm. The difference between the CP2k and the other two schemes
increases with decreasing raindrop radius; especially when the raindrop radius is





smaller than 50 μm, with the maximum difference being more than two orders of
magnitude. Therefore, the probability density distributions (PDFs) of raindrop radius
are important for the difference between different accretion rate schemes. Figure 10
shows the probability density distributions (PDFs) of raindrop radius used in the
accretion process in the three schemes. All raindrops are smaller than $10^3$ μm. The PDFs
have peaks of ~30, ~30 and ~25 μm in the control run, the Ko13 and the CP2k,
respectively, and the cumulative PDF shows that the raindrops with radius smaller than
50 μm have frequencies of 58.8%, 53.8%, and 46.0%, respectively. The drop size
distributions from both aircraft observations and bin models confirm that a large
proportion of liquid droplets have radii larger than 25 μm but smaller than 50 μm (Wood,
2005b; Morrison and Grabowski, 2007). Such large percentage of small raindrops
makes the accretion rate and precipitation in the CP2k quite different from that in other
schemes (Figure 9). Furthermore, there is a positive feedback mechanism, since
accretion increases $q_r$ and accretion rate is positively correlated with $q_r$. The
overestimation of the accretion rate in the control run hence feeds back on itself. This
is the reason why the precipitation and accretion rate differences between the control
run and the CP2k are so different over the southeastern corner in domain 02 and during
the second peak period in domain 03.
Previous studies have shown that, to initiate liquid phase precipitation, the cloud
effective radius needs to reach about 14 μm (Rosenfeld et al., 2019). A closer look on
the cloud droplet size distributions is hence informative to understand the differences





in precipitation behavior between the CP2k and the other experiments. Figure 11 shows
the liquid-phase precipitation rate as a function of cloud droplet effective radius. The
liquid-phase precipitation rate is estimated as the product of total precipitation and the
ratio of liquid-phase process rates (autoconversion + accretion) and ice-phase process
rates (melting from snow + graupel). The liquid-phase precipitation rate exceeds 2
mm/day when the cloud effective radius is 9 μm in the control run and the Ko13. In the
CP2k, it is not until the cloud effective radius reaches about 15 μm, that the precipitation
rate exceeds 2 mm/day. The contribution from autoconversion is close to 0 in the control
run, which could be due to the consumption of cloud droplets by accretion after droplets
reach 9 μm. The value of 9 μm, is much smaller than 14 μm needed to initiate liquid-
phase precipitation, often suggested by observational studies. On the contrary, there is
a significant increase in liquid-phase precipitation rate from the autoconversion process
in the CP2k at 15 μm and then the accretion process begins to efficiently produce liquid-
phase precipitation. Therefore, the improvement in the CP2k surface precipitation
compared to the control, appears to occur for the right reasons.

**4. Summary and conclusions**
In this paper, a typical summer plateau precipitation event over the Tibetan Plateau
is simulated using the WRFv3.8.1 model with the Morrison double-moment scheme.
The control run reproduces the primary spatial distribution and temporal evolution of
precipitation rate. However, the precipitation in the coarse resolution domain is about





twice of the observed value, similar to previous studies which claimed that the
overprediction was due to low resolution or inaccurate large-scale forcing. The
precipitation in the higher resolution domain is more consistent with the observations,
but still, the precipitation during the second precipitation peak period in this domain is
overpredicted.

To understand the roles of liquid-phase microphysical processes in the

overprediction of precipitation, sensitivity tests are carried out by introducing different
parameterizations of liquid-phase processes into the Morrison double-moment scheme,
including three autoconversion parameterizations (Be68, Bh94 and LD04), two
accretion parameterizations (CP2k and Ko13), and one entrainment-mixing
parameterization (INHOMO).

The overprediction of precipitation is significantly reduced in both the low- and

high-resolution domains in the experiment using the Cohard and Pinty (2000) accretion
scheme (CP2k). The Heidke skill scores with the CP2k also show better results
compared to other cases. Furthermore, each simulation is further divided into two parts:
one with dominant ice-phase processes, the other with dominant liquid-phase processes.
The simulations have the largest differences when the liquid-phase processes dominate,
and the improvement in the CP2k experiment is more pronounced in this case. When
the ice-phase processes are important, all the simulations are equivalent, including the
CP2k. There are several reasons for this behavior. The accretion rate is smaller in the
CP2k experiment than that in the control run, which suppresses precipitation due to





liquid-phase processes. Due to weaker accretion, more cloud droplets remain suspended
in the atmosphere and are available for riming onto snow and graupel. Precipitation due
to melting from snow and graupel is then enhanced. The combination of the weaker
accretion and stronger melting in the CP2k offset each other. That is the reason why the
precipitation does not change much in the CP2k when ice-phase processes dominate.
When the ice-phase processes are relatively weak, the precipitation from the enhanced
riming and melting processes cannot compensate the loss of precipitation due to the
suppression of accretion. Therefore, the precipitation rate is smaller in the CP2k than
in the control run.

To understand the physical reasons for the improved performance of the CP2k, the

equations for parameterizing the accretion rate in the CP2k, the KK and the Ko13 are
compared directly. The accretion rate in the CP2k is always smaller than in the KK or
Ko13 scheme when the raindrop radius is smaller than 2000 μm. Furthermore, the
difference increases with decreasing raindrop radius and can amount to more than two
orders of magnitude when the raindrop radius is smaller than 50 μm. The PDFs of
raindrop radii have their peaks around 30 μm. Around 50% of raindrops have radius
less than 50 μm. This is the reason why the CP2k suppresses accretion and liquid-phase
precipitation compared to the other two schemes. Further insight in the reasons for
different behavior in the CP2k compared to the other schemes is provided through the
relation of cloud droplet size and liquid phase precipitation rates. It is often claimed
that, to initiate liquid-phase precipitation, cloud effective radius needs to reach 14 μm.





When the cloud effective radius is 9 μm in the control run and the Ko13, the liquid-
phase precipitation rate already exceeds 2 mm/day however; In the CP2k, on the other
hand, liquid phase precipitation does not start until the effective radius reaches about
15 μm, which is more consistent with observations.

**Author contributions.** CL and XX designed the experiments. XX carried out the
experiments and conducted the data analysis with contributions from all coauthors.
KVW developed the model code. XX prepared the paper with help from CL, YL, WG,
YW, YC, SL, and KVW.

**Competing interests.** The authors declare that they have no conflict of interest.

**Acknowledgements.** The authors thank the Amy Solomon for providing the
microphysics scheme, Yinzhao Ma and Yang Hong for providing the precipitation data.
This research is supported by the National Key Research and Development Program of
China (2017YFA0604000), the Natural Science Foundation of Jiangsu Province
(BK20160041), the National Natural Science Foundation of China (41822504,
91537108), the Qinglan Project (R2018Q05), and the Six Talent Peak Project in Jiangsu
(2015-JY-011). Liu is supported by the U.S. Department of Energy Office of Science
Biological and Environmental Research as part of the Atmospheric Systems Research
(ASR) Program and Solar Energy and Technology Office (SETO). Brookhaven





National Laboratory is operated by Battelle for the U.S. Department of Energy under
Contract DE-SC00112704.

**Appendix A: Symbol List**
$N_c$: number concentration of cloud droplets
$q_c$: mixing ratio of cloud droplet
$N_r$: number concentration of raindrops
$q_r$: mixing ratio of raindrops
$N_i$: number concentration of ice crystals
$q_i$: mixing ratio of ice crystals
$N_s$: number concentration of snow particles
$q_s$: mixing ratio of snow particles
$N_g$: number concentration of graupel particles
$q_g$: mixing ratio of graupel particles
$R_{accr}$: conversion rate of accretion process
$R_{auto}$: conversion rate of autoconversion process
$\rho_a$: air density
$\varepsilon$: dispersion
$\rho_W$: water density
$\lambda$: slope parameter
$N_{c0}$: number concentration of cloud water droplets before evaporation process



$q_{c0}$: mixing ratio of cloud water droplets before evaporation process
$p_t$: the threshold value of precipitation in the Heidke skill score
$p_s$: value of precipitation from simulations in the Heidke skill score
$p_o$: value of precipitation from observation in the Heidke skill score
$\tau$: cloud optical depth
$\overline{r_e}$: averaged effective radius of cloud water droplets
LCWP: liquid cloud water path
EVAP-r: evaporation of raindrops
ACCR-r: accretion of cloud liquid water by rain
AUTO-r: autoconversion from cloud droplets to raindrops
MELT: melting from snow or graupel particles to raindrops
AUTO-s: autoconversion of cloud ice to snow
ACCR-s: accretion of cloud ice by snow
RIM-s: accretion of cloud droplets by snow particle
RIM-g: accretion of cloud droplets by graupel particle






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



**Caption List:**
**Table 1.** Contingency table used to calculate the Heidke skill score (HSS). The elements
*a-d* represent the numbers of "hits", "false alarms", "misses" and "correct negatives",
respectively. $p_t$ is the threshold value of precipitation in observation and simulations, $p_s$
is the value from simulations and $p_o$ is the value from observations.
**Table 2.** The values of four elements *a-d* and Heidke skill score (HSS) for all
simulations over domain 02 and domain 03 (d02/d03), respectively.
**Table 3.** The mean number concentration $N_c$ (/cm$^3$), effective radius $\overline{r_e}$ (μm) of cloud
droplets, area-averaged liquid cloud water path LCWP (g/m$^2$), cloud optical depth $\tau$
over domain 02 and 03 (d02/d03) of the control run, Be68, Bh94, LD04 (different
autoconversion schemes), CP2k, Ko13 (different accretion schemes) and INHOMO run
(different mixing mechanism).
**Figure 1.** Geographic locations of the three domains used in the numerical simulation.
**Figure 2.** Spatial distributions of 48 h accumulated precipitation (mm) during 0000
UTC 22 July to 0000 UTC 24 July 2014 from the observations and the control run over
domain 02 (a, b) and domain 03 (c, d).
**Figure 3.** Time series of area-averaged hourly precipitation rate (mm/h) during 0000
UTC 22 July to 0000 UTC 24 July 2014 over (a) domain 02 and (b) domain 03 from
the observations and the control run.
**Figure 4.** Mean vertical profiles of mixing ratios (g/kg) of cloud droplets ($q_c$),
raindrops($q_r$), ice particles($q_i$), snow particles ($q_s$), graupel particles($q_g$) and their



primary microphysical processes in the control run (a, b) averaged from 48 h over
domain 02 except southeastern corner, (c, d)averaged from 48 h at southeastern corner
over domain 02, averaged during two precipitation peaks (e, f) 0700-1200 UTC 22 July
2014 and (g, h) 0700-1200 UTC 23 July 2014 over domain 03. The purple dot-dash
lines denote the mean height of 0 ℃ isotherm.
**Figure 5.** Spatial distributions of 48 h accumulated precipitation (mm) during 0000
UTC 22 July to 0000 UTC 24 July 2014 from observations and all sensitivity
simulations over (a-f) domain 02 and (g-l) domain 03.
**Figure 6.** Time series of area-averaged hourly precipitation rate (mm/h) during 0000
UTC 22 July to 0000 UTC 24 July 2014 over (a) domain 02 and (b) domain 03 from
the observations and all simulations.
**Figure 7.** The time series of area-averaged autoconversion rate and accretion rate over
(a, b) domain 02 and (c, d) domain 03 for all simulations, respectively.
**Figure 8.** Differences of mean vertical profiles of the dominated microphysical
processes conversion rates between the CP2k and the control run (CP2k-Control) from
(a) domain 02 except southeastern corner, (b) the southeastern corner of domain 02, and
during the two precipitation peak periods (c) 0700-1200 UTC 22 July and (d) 0700-
1200 UTC 23 July over domain 03. The purple dot-dash lines denote the mean height
of 0 ℃ isotherm.





**Figure 9.** The accretion rate as a function of raindrop radius with fixed cloud mixing
ratio $q_c$ = 1 g/kg, the radius of cloud droplet $R_c$ = 10 μm, number concentration of
raindrops $N_r$ = 4000 /m³ for the three accretion schemes.
**Figure 10.** Probability distribution function (PDF) and cumulative PDF of raindrop
radius involved in the accretion process for (a) the control run, (b) the CP2k, and (c)
the Ko13. The purple line denotes the radius of raindrop equal to 50 μm.
**Figure 11.** Dependence of warm rain intensity on cloud effective radius from the
control run and the CP2k during 0000 UTC 22 July to 0000 UTC 24 July 2014 over
domain 03.





Table 1. Contingency table used to calculate the Heidke skill score (HSS). The elements
*a-d* represent the numbers of "hits", "false alarms", "misses" and "correct negatives",
respectively. $p_t$ is the threshold value of precipitation in observation and simulations, $p_s$
is the value from simulations and $p_o$ is the value from observations.

|  | Observation $p_o > p_t$ | Observation $p_o \leq p_t$ |
|---|---|---|
| Simulation $p_s > p_t$ | *a* | *b* |
| Simulation $p_s \leq p_t$ | *c* | *d* |







Table 2. The values of four elements *a-d* and Heidke skill score (HSS) for all
simulations over domain 02 and domain 03 (d02/d03), respectively.

|  | *a* | *b* | *c* | *d* | HSS |
|---|---|---|---|---|---|
| ***control*** | 2636/304 | 1224/148 | 773/76 | 2231/48 | 0.419/0.049 |
| ***autoconversion*** | | | | | |
| Be68 | 2645/309 | 1261/142 | 764/71 | 2194/54 | 0.411/0.097 |
| Bh94 | 2533/306 | 1148/138 | 876/74 | 2307/58 | 0.411/0.110 |
| LD04 | 2628/313 | 1264/154 | 781/67 | 2191/42 | 0.405/0.043 |
| ***accretion*** | | | | | |
| CP2k | 2583/304 | 1063/129 | 632/76 | 2586/67 | 0.508/0.152 |
| K013 | 2620/303 | 1223/146 | 770/77 | 2251/50 | 0.420/0.057 |
| ***mixing mechanism*** | | | | | |
| INHOMO | 2656/308 | 1124/141 | 753/72 | 2214/55 | 0.420/0.100 |







Table 3. The mean number concentration $N_c$ (/cm$^3$), effective radius $\overline{r_e}$ (µm) of cloud
droplets, area-averaged liquid cloud water path LCWP (g/m$^2$), cloud optical depth $\tau$
over domain 02 and 03 (d02/d03) of the control run, Be68, Bh94, LD04 (different
autoconversion schemes), CP2k, Ko13 (different accretion schemes) and INHOMO run
(different mixing mechanism).

| | $N_c$(/cm$^3$) | LCWP(g/m$^2$) | $\tau$ | $\overline{r_e}$ (µm) |
|---|---|---|---|---|
| ***control*** | 71.5/91.2 | 73.5/66.8 | 11.9/11.1 | 6.97/6.77 |
| ***autoconversion*** | | | | |
| Be68 | 71.3/91.6 | 63.4/59.4 | 10.8/10.6 | 6.84/6.74 |
| Bh94 | 72.3/91.9 | 81.3/76.1 | 12.7/12.6 | 7.13/7.01 |
| LD04 | 71.6/91.3 | 63.8/60.9 | 10.8/10.6 | 6.85/6.69 |
| ***accretion*** | | | | |
| CP2k | 72.4/90.1 | 121.0/97.0 | 16.3/14.3 | 7.52/7.15 |
| Ko13 | 71.5/91.0 | 74.4/64.4 | 11.6/10.9 | 6.92/6.72 |
| ***mixing mechanism*** | | | | |
| INHOMO | 68.9/86.3 | 72.9/66.7 | 11.7/11.0 | 7.03/6.87 |







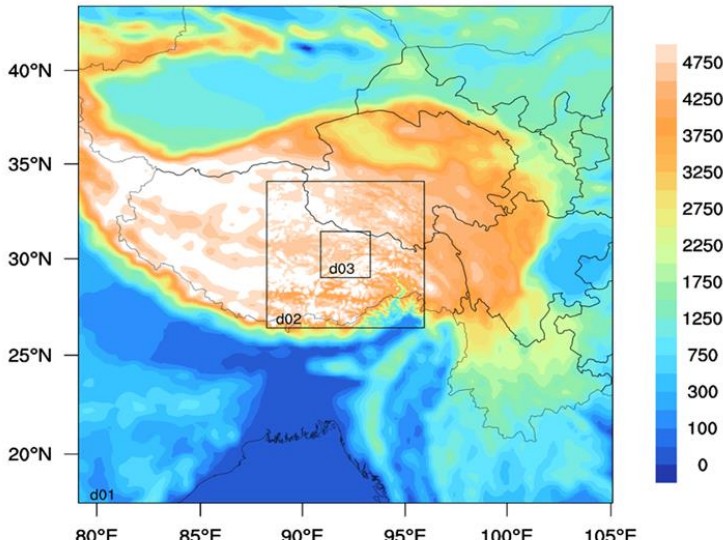


Figure 1. Geographic locations of the three domains used in the numerical simulation.






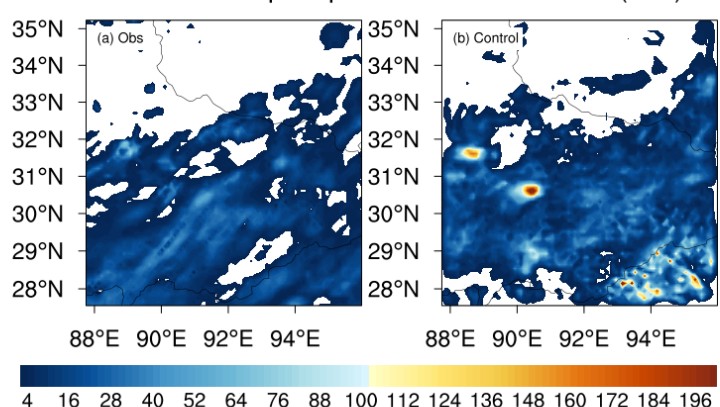

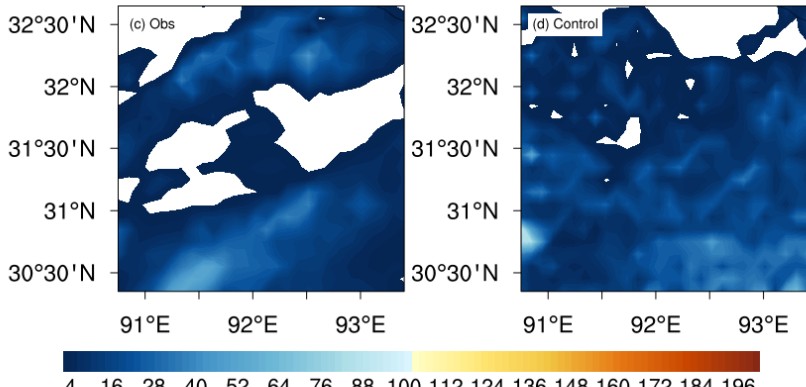


Figure 2. Spatial distributions of 48 h accumulated precipitation (mm) during 0000
UTC 22 July to 0000 UTC 24 July 2014 from the observations and the control run over
domain 02 (a, b) and domain 03 (c, d).





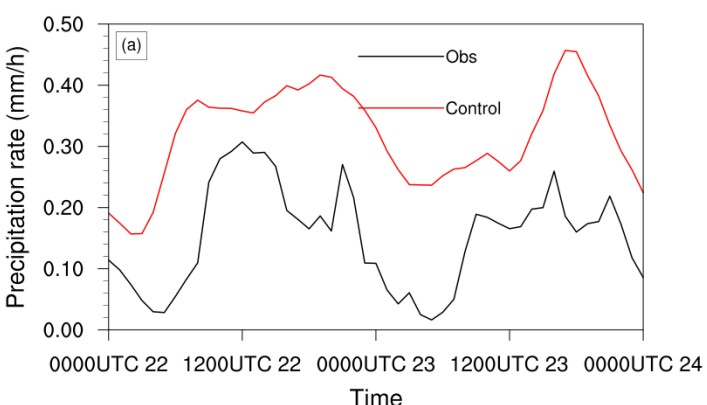

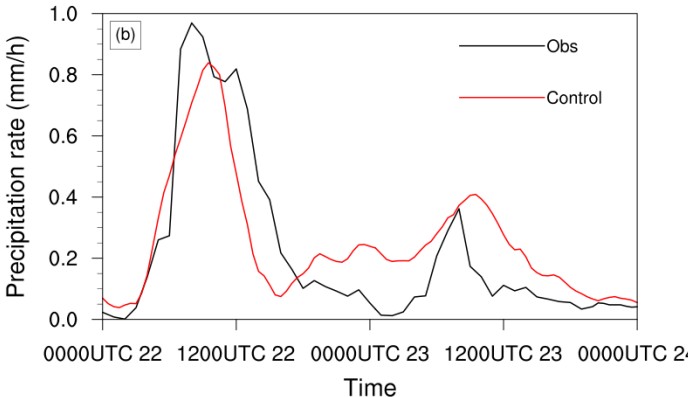


Figure 3. Time series of area-averaged hourly precipitation rate (mm/h) during 0000
UTC 22 July to 0000 UTC 24 July 2014 over (a) domain 02 and (b) domain 03 from
the observations and the control run.

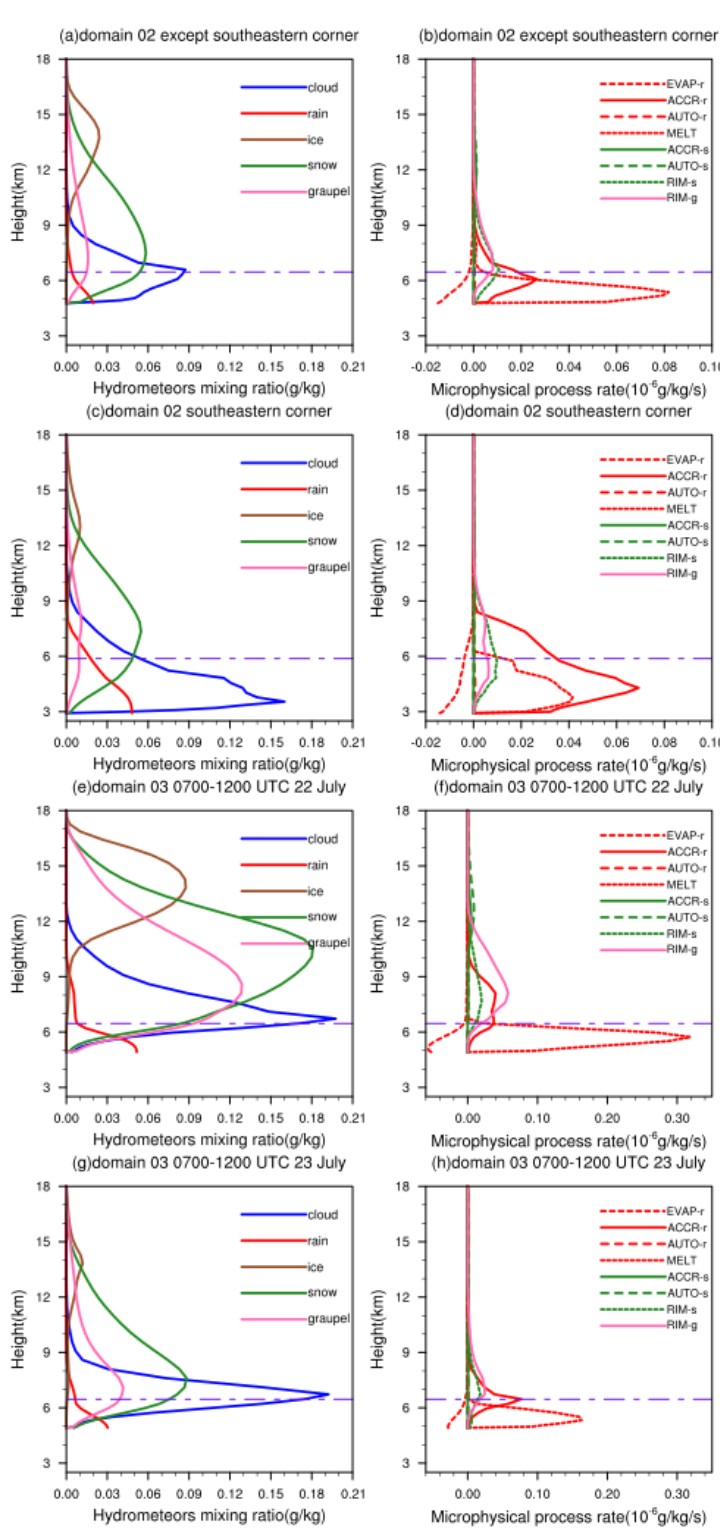



Figure 4. Mean vertical profiles of mixing ratios (g/kg) of cloud droplets ($q_c$),
raindrops($q_r$), ice particles($q_i$), snow particles ($q_s$), graupel particles($q_g$) and their
primary microphysical processes in the control run (a, b) averaged from 48 h over
domain 02 except southeastern corner, (c, d)averaged from 48 h at southeastern corner
over domain 02, averaged during two precipitation peaks (e, f) 0700-1200 UTC 22 July
2014 and (g, h) 0700-1200 UTC 23 July 2014 over domain 03. The purple dot-dash
lines denote the mean height of 0 ℃ isotherm. The meanings of the symbols in the
legends are shown in Appendix A.

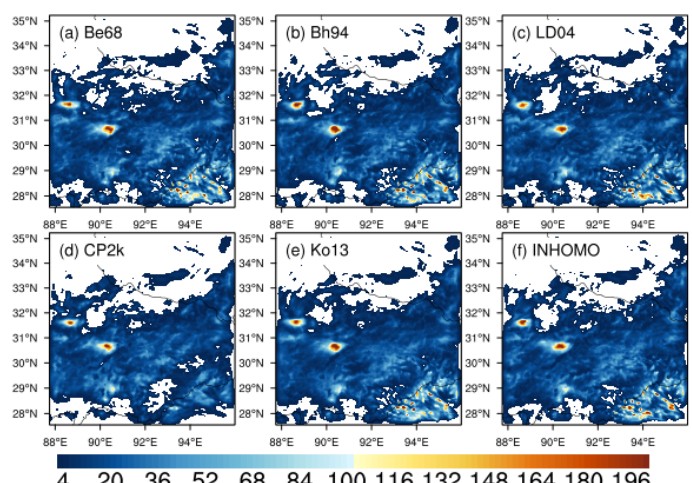

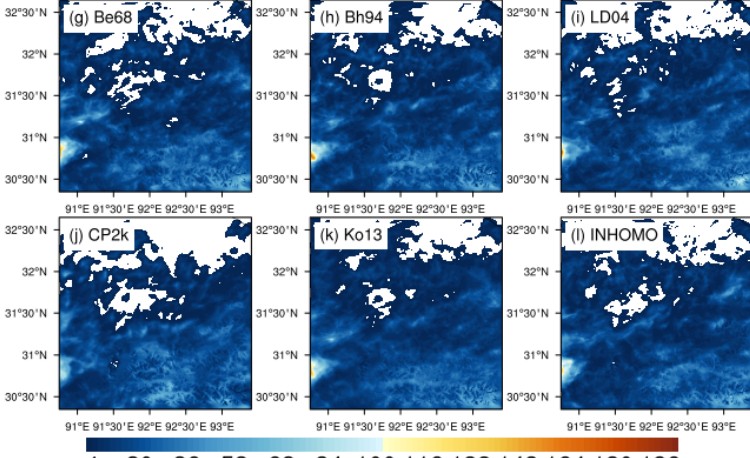


Figure 5. Spatial distributions of 48 h accumulated precipitation (mm) during 0000
UTC 22 July to 0000 UTC 24 July 2014 from observations and all sensitivity
simulations over (a-f) domain 02 and (g-l) domain 03.



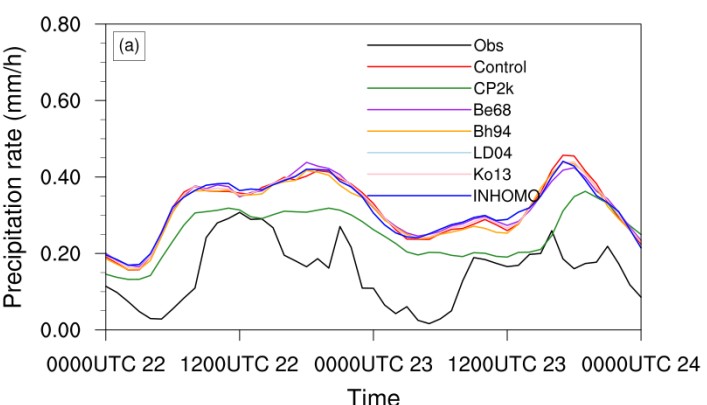

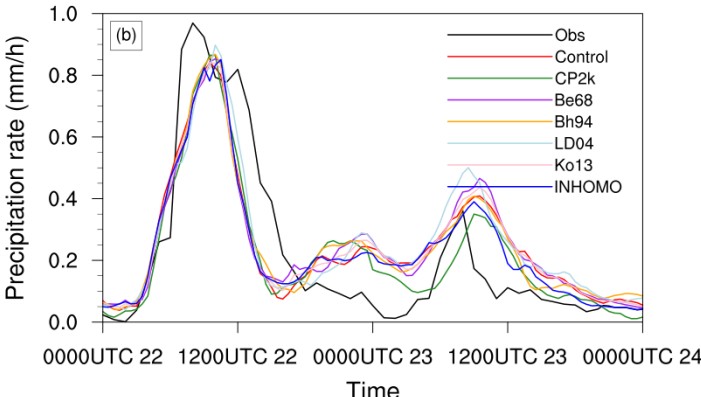


Figure 6. Time series of area-averaged hourly precipitation rate (mm/h) during 0000
UTC 22 July to 0000 UTC 24 July 2014 over (a) domain 02 and (b) domain 03 from
the observations and all simulations.





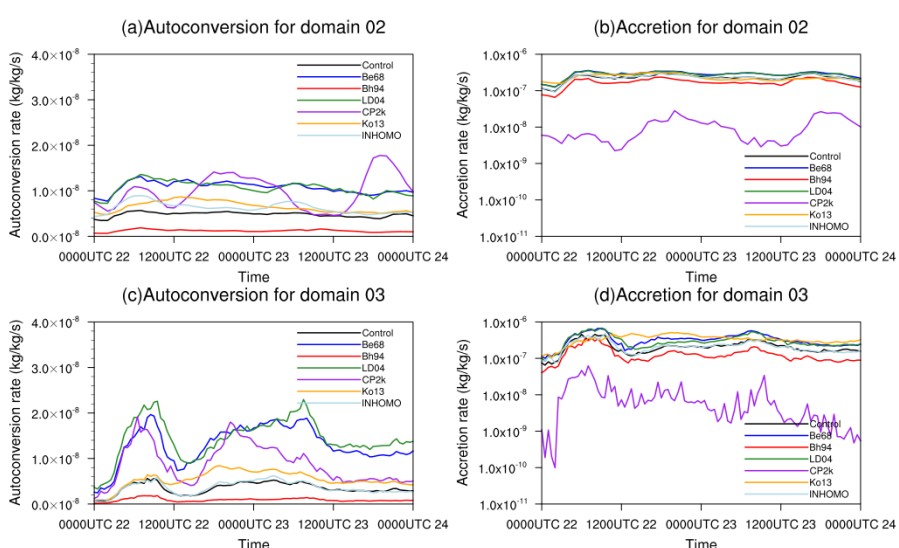


Figure 7. The time series of area-averaged autoconversion rate and accretion rate over
(a, b) domain 02 and (c, d) domain 03 for all simulations, respectively.



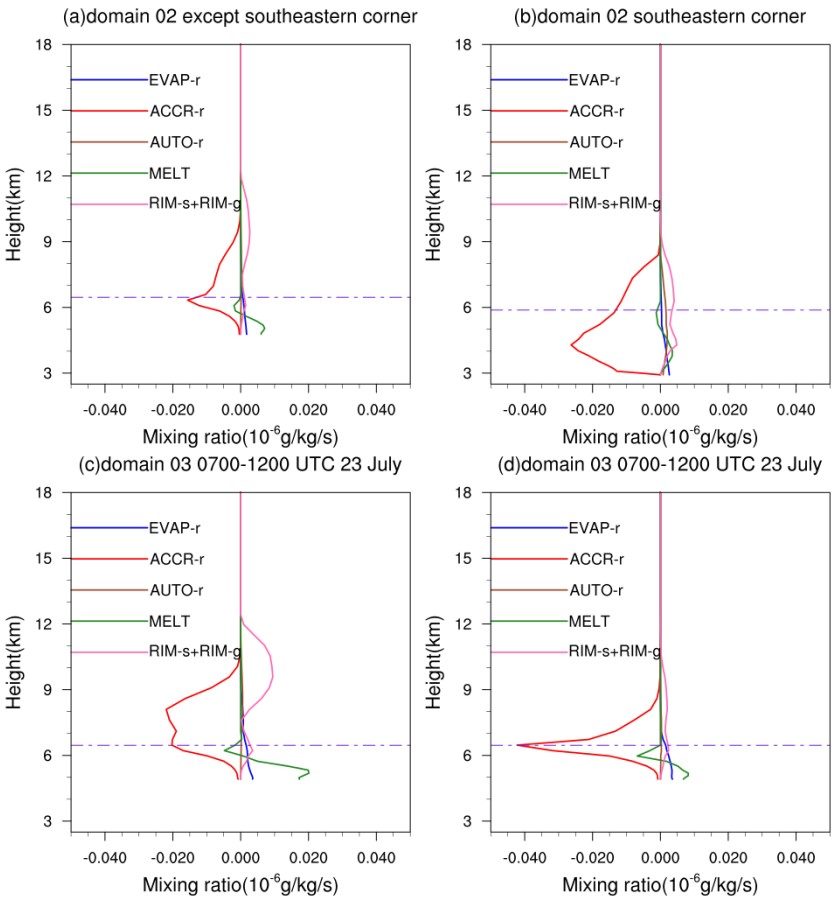

Figure 8. Differences of mean vertical profiles of the dominated microphysical

processes conversion rates between the CP2k and the control run (CP2k-Control) from

(a) domain 02 except southeastern corner, (b) the southeastern corner of domain 02, and

during the two precipitation peak periods (c) 0700-1200 UTC 22 July and (d) 0700-

1200 UTC 23 July over domain 03. The purple dot-dash lines denote the mean height

of 0 ℃ isotherm. The meanings of the symbols in the legends are shown in Appendix

A.





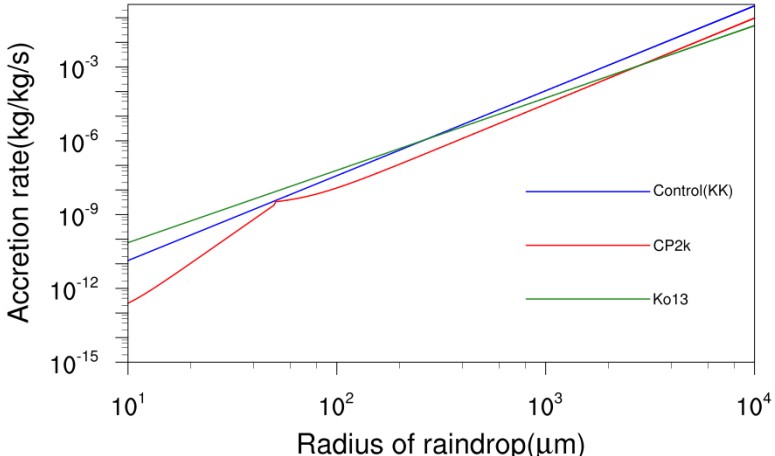


Figure 9. The accretion rate as a function of raindrop radius with fixed cloud mixing
ratio $q_c$ = 1 g/kg, the radius of cloud droplet $R_c$ = 10 μm, number concentration of
raindrops $N_r$ = 4000 /m$^3$ for the three accretion schemes.



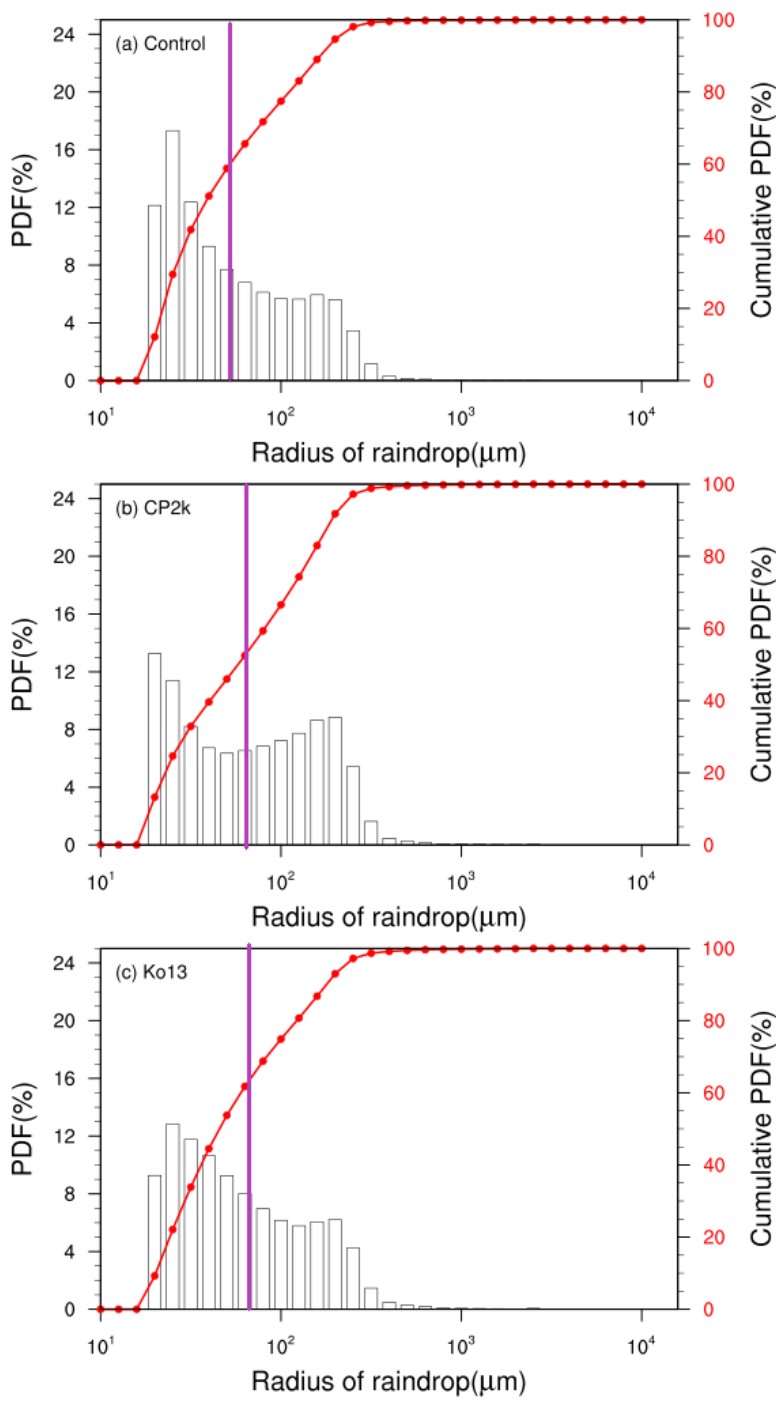


Figure 10. Probability distribution function (PDF) and cumulative PDF of raindrop



radius involved in the accretion process for (a) the control run, (b) the CP2k, and (c)
the Ko13. The purple line denotes the radius of raindrop equal to 50 μm.

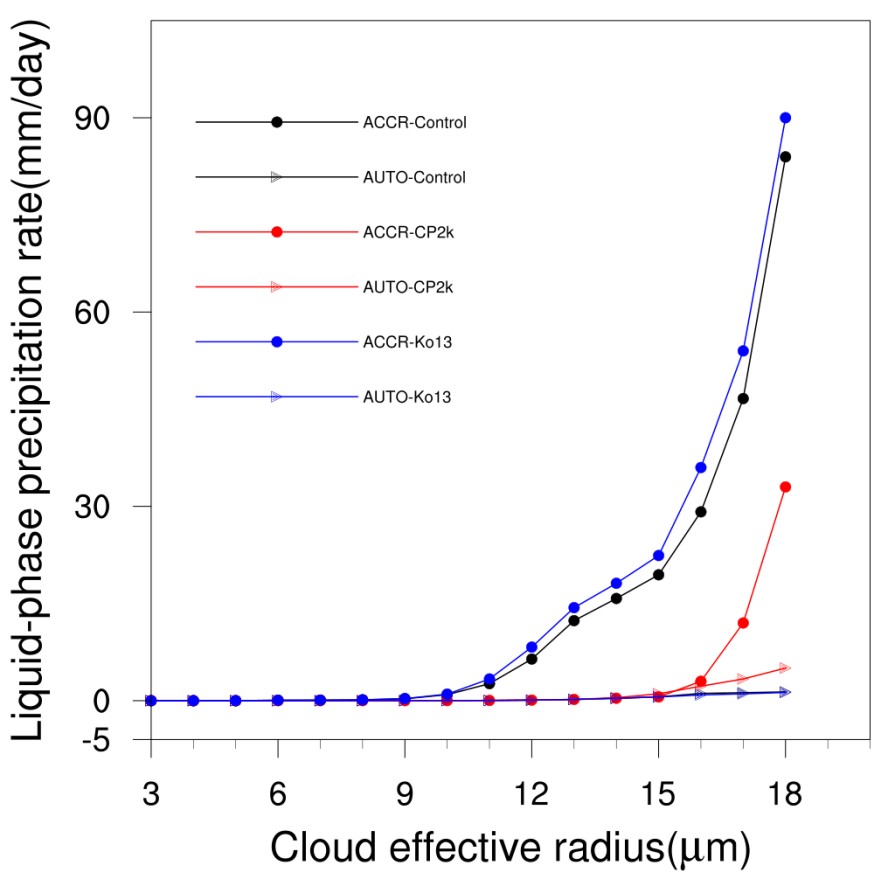


Figure 11. Dependence of liquid-phase precipitation intensity on cloud effective radius
from the three accretion schemes during 0000 UTC 22 July to 0000 UTC 24 July 2014
over domain 03.