# Peer review of "Effects of Liquid Phase Cloud Microphysical Processes in Mixed"

_Atmospheric Chemistry and Physics, 2019_

## Referee Comment (RC1) · Anonymous Referee #1 · 26 Dec 2019

Review for Xiaoqi Xu et al. "Effects of Liquid Phase Cloud Microphysical Processes in Mixed Phase Cumulus Clouds over the Tibetan Plateau"

General comments:

The authors investigated the effects of liquid microphysics scheme (e.g., autoconversion and accretion) on the surface precipitation rate over the Tibetan Plateau (TP) using Weather Research and Forecasting (WRF) model with the double-moment Morrison scheme. Although the authors stated that the impact of entrainment mixing was also explored, I did not find enough discussion and supporting data.

In the beginning, the authors addressed that the model resolution is critical for predicting accurate precipitation over the TP region, but the main body of the manuscript looked at the different treatment of liquid microphysics scheme assumed in the model. I unfortunately do not understand the main purpose of this study because it is still unclear why the authors chose the TP region to investigate uncertainty in the microphysical scheme. I strongly suggest reframing the overall structure, in particular the introduction section at least, to highlight the study goal, motivation, and new findings of this study.

It is an interesting result that the accretion scheme which depends on raindrop size (Cohard and Pinty, 2000) as well as cloud and rain mass mixing ratios (qc and qr) performs better than the other standard accretion schemes which depend only on qc and qr. Although the authors present some interesting results with the scheme comparison, almost findings remained within the scope of the previous studies based on theoretical modeling (e.g., Wood, 2005; Wood et al., 2009; Lee and Baik, 2017) and climate modeling (e.g., Gettelman et al., 2013, 2015). I feel that the authors should add more discussion about the physical mechanism behind the different scheme used.

Therefore, I suggest major revisions and re-review. The authors also have presentation issues to address detailed below.

Specific comments

Lines 55-56 and 59-63: Is it true for the WRF model? Again, is this issue attributed to the model resolution? Or microphysical parameterization?

Line 82-84: This statement is too vague.

Line 86: Where did you state the entrainment-mixing issue? I did not find. This should be removed unless the authors provide data and discussion.

Line 94-95: Please add reference(s) here.

Line 120: Which satellite data did you use? Please add the description and appropriate citation(s).

Line 148-149, Equations (1) and (2): The unit for mass tendency is kg/kg/s in general. Equation (5): Please define xc.

Line 180: "most accretion schemes" needs a couple of references.

Line 219-222: It is unclear because a lower precipitation rate is smoothed by blue. The color scheme in Figure 2 (and also Figure 5) should be modified. In the present figure, I cannot recognize the "rainband oriented in the northeast-southwest direction".

Lines 347-349 and 353-355, and Figures 7b and 7d: I disagree with this sentence. The autoconversion rate in the CP2k scheme is higher (Figures 7a and 7c), whereas the accretion rate is the lowest. This is inconsistent with your explanation. And also, why the accretion rates are almost similar among various schemes except for the CP2k? If the same accretion scheme (KK00) is used except for the CP2k, it is natural that the accretion rates are almost the same among the other schemes. I am confused about that, and please describe a more detailed explanation.

Sections 3.2.2.1, 3.2.2.2, and 3.2.2.3: These sections should include more discussion. The current description only lists the result from different microphysical scheme use, but the variability due to the different treatment of autoconversion and accretion schemes has been well known (e.g., Wood, 2005; Wood et al., 2009; Gettelman et al., 2013, 2015; Jing et al., 2019). It would help readers if more in-depth discussions with relevant studies are shown in the text.

Line 367-368: The difference of Nc between the INHOMO run and CTRL run is quite small. It is better to conclude that the impact of the entrainment-mixing is less important than the liquid conversion process in this study, although it depends strongly on the cloud type simulated in the TP area. Please add more discussion here (or conclusion section).

Line 400-404: This may imply that the uncertainties in ice- and mixed-phase microphysics schemes are conveyed to the uncertainty of surface precipitation. To confirm

this, an additional simulation without the riming process will be helpful to understand the microphysical mechanisms.

Line 424-426: Does it mean that the other microphysical schemes except for the CP2k overestimate the accretion rate and consequent precipitation? Or is this the case only for the TP region? I suggest that the authors refer to the ratio of accretion over autoconversion (e.g., Gettelman et al., 2013; Seifert and Onishi, 2016; Lee and Baik, 2017) among the experiments. This metric will be helpful to evaluate the dependence of microphysical process rates on cloud regimes in the model.

Lines 483-485 and 491-492: Why does the CP2k scheme improve the too early onset of precipitation in spite of the higher autoconversion rate compared with the other schemes?

Figure 1: Please explain the color shade.

Figures 2 and 5: Unclear color contrast. Please consider to change the current color scheme (blue-to-red) to standard rainbow color or white-to-blue etc.

References

Cohard and Pinty (2000): A comprehensive two-moment warm microphysical bulk scheme. I: Description and tests, doi:10.1002/qj.49712656613

Gettelman et al. (2013): Microphysical process rates and global aerosol–cloud interactions, doi:10.5194/acp-13-9855-2013

Gettelman et al. (2015): Advanced two-moment bulk microphysics for global models. Part II: Global model solutions and aerosol–cloud interactions, doi:10.1175/JCLI-D-14-00103.1

Lee and Baik (2017): A physically based autoconversion parameterization, doi:10.1175/JAS-D-16-0207.1

Seifert and Onishi (2016): Turbulence effects on warm-rain formation in precipitating

shallow convection revisited, doi:10.5194/acp-16-12127-2016

Wood (2005): Drizzle in stratiform boundary layer clouds. Part II: Microphysical aspects, doi:10.1175/JAS3530.1

Wood et al. (2009): Understanding the importance of microphysics and macrophysics for warm rain in marine low clouds. Part II: Heuristic models of rain formation, doi:10.1175/2009JAS3072.1

---

## Referee Comment (RC2) · Anonymous Referee #2 · 4 Feb 2020

I have reviewed "Effects of Liquid Phase Cloud Microphysical Processes in Mixed Phase Cumulus Clouds over the Tibetan Plateau" by Xu et al. I do not have any complaints about the analysis itself, but I have serious doubts that the manuscript is sufficiently relevant beyond the one case investigated to be within the scope of ACP.

The article describes an analysis performed on a single synoptic system transiting the Tibetan Plateau. The authors focus on warm cloud processes, performing sensitivity studies with different autoconversion/accretion/droplet evaporation parameterizations; why they focus on these processes is not well explained, in particular since the precipitation in their study case is clearly initiated in the ice phase (figure 4), so one would

expect that only accretion and ice/mixed-phase processes matter.

Not surprisingly, the authors find that autoconversion and homogeneous vs inhomogeneous cloud droplet evaporation make very little difference in accumulated precipitation. In a revised manuscript, I would suggest getting rid of several pages of unsurprising results and replacing them simply with one paragraph along the lines of, "we analyzed the effect of different autoconversion parameterizations and mixing assumptions and found them to have no substantial impact."

The finding that accretion is an important control on accumulated precipitation is also not very surprising. Furthermore, it is not clear what we are supposed to do with this information. When parameterizations are developed, they are usually tuned to do something reasonable in one or a handful of test cases, but it is understood that they will probably not give results that match observations in every conceivable case – usually far from it. So it is not surprising that some parameterizations do better than others at reproducing this particular case. However, that does not mean that the winner in this case will produce the best results in other cases. Are the authors recommending that the Cohard and Pinty (2000) accretion parameterization should be used generally, or generally for Tibetan Plateau studies? How does one case study support that recommendation? If that is not the recommendation, what is new or useful about the results? That different warm cloud microphysics schemes can lead to wildly different simulations of individual cases is nothing new; for an example of a study that draws this conclusion in a more generalized way, with interesting statements about science implications, see White et al. (2017), https://doi.org/10.5194/acp-17-12145-2017

Thus, I recommend that the authors substantially revise their manuscript to focus on conclusions that are of use beyond this one case study. If this is not possible, I do not think that ACP is the appropriate journal.
* * *
[Figure]

2019.

---

## Author Comment (AC1) · 25 Mar 2020

**Response to Anonymous Referee #1**

Review for Xiaoqi Xu et al. "Effects of Liquid Phase Cloud Microphysical Processes in Mixed Phase Cumulus Clouds over the Tibetan Plateau"

General comments:

**1.** The authors investigated the effects of liquid microphysics scheme (e.g., autoconversion and accretion) on the surface precipitation rate over the Tibetan Plateau (TP) using Weather Research and Forecasting (WRF) model with the double-moment Morrison scheme. Although the authors stated that the impact of entrainment mixing was also explored, I did not find enough discussion and supporting data.

Reply: Thank you for your comments. A sensitivity test is run assuming extremely inhomogeneous mixing mechanism (INHOMO in Table 3). For the entrainment-mixing processes, the INHOMO experiment has the largest effect on cloud number concentration ($N_c$) among all the sensitivity tests. Even so, there is only a modest reduction of only 2.6 (4.9) /cm$^3$ compared to the control run, resulting in 0.9% (1.5%) larger cloud effective radius ($\overline{r_e}$) over domain 02 (03). Such a variation of $\overline{r_e}$ over domain 03 is comparable with that in all the autoconversion schemes and Ko13. The variation of liquid cloud water path (LCWP) is smaller than that in the liquid conversion process. (Page 18, Line 359-364).

Besides the INHOMO case, we have added new sensitivity tests in the revised manuscript. Sensitivity tests of all autoconversion and accretion schemes in Table 3 are conducted assuming different entrainment-mixing mechanisms. The impacts of different entrainment-mixing mechanisms are still small. Over domain 02, the differences of LCWP, $\overline{r_e}$ and $N_c$ due to different entrainment-mixing mechanisms are, respectively, in the ranges of 0.1%-0.9%, 0.7%-0.9% and 3.2%-3.9%. Over domain 03, the differences are, respectively, in the ranges of 0%-1.7%, 1.5%-2.0% and 5.3%-6.9% (Supplement, Page 1-2, Line 9-22).

Combined with the other referee comment, we have added discussions in the main text (Page 18, Line 364-369) and also in the supplement (Page 1-2, Line 9-22).

**2.** In the beginning, the authors addressed that the model resolution is critical for predicting accurate precipitation over the TP region, but the main body of the manuscript looked at the different treatment of liquid microphysics scheme assumed in the model. I unfortunately do not understand the main purpose of this study because it is still unlear why the authors chose the TP region to investigate uncertainty in the microphysical scheme. I strongly suggest reframing the overall structure, in particular the introduction section at least, to highlight the study goal, motivation, and new findings of this study.

Reply: Sorry for the unclear organization. The main purpose of our study is to investigate how to mitigate the overprediction of precipitation over the TP region and what are the roles of the different liquid-phase microphysical processes. As discussed in the introduction (Page 5, Line 82-91): it is still unknown how liquid-phase processes affect precipitation over the TP, and whether improving the parameterizations of liquid-phase processes can mitigate the problem of overpredicted precipitation. Also unknown is which liquid-phase process is the most important in affecting TP precipitation, and which parameterization can best describe the most important process and why. We also confirm that the model resolution is another reason responsible for precipitation overprediction, as claimed in many previous studies (e.g., Sato et al., 2008, Xu et al., 2012).

We have reframed the structures of Section 1 Introduction and Section 3 Case study Analysis. All paragraphs related to model resolution's effects on precipitation overprediction are moved to the "Section 5 Sensitivity to horizontal resolution (Page 26-28)", including the literature review in the introduction, and different resolutions' effects on precipitation in domains 02 and 03 in Section 3.1.1. Correspondingly, the conclusion section and abstract are also reframed (Page 30, Line 617-621; Page 2, Line 29-31). Besides, to highlight the main purpose of this study, the phase "precipitation overprediction" is emphasized in all sections in the revised manuscript. For example, the titles of Sections 3.2 and 3.3 are modified to be "Sensitivity of precipitation overprediction to different liquid-phase processes" and "Reasons for improvements of precipitation overprediction in CP2k", respectively.

References

Sato, T., Yoshikane, T., Satoh, M., Miura H., and Fujinami, H.: Resolution dependency of the diurnal cycle of convective clouds over the Tibetan Plateau in a mesoscale model, Journal of the Meteorological Society of Japan. Ser. II, 86, 17-31, 2008.

Xu, J., Zhang, B., Wang, M., and Wang, H.: Diurnal variation of summer precipitation over the Tibetan Plateau: a cloud-resolving simulation, Annales Geophysicae, 2012, 1575-1586,

**3.** It is an interesting result that the accretion scheme which depends on raindrop size (Cohard and Pinty, 2000) as well as cloud and rain mass mixing ratios (qc and qr) performs better than the other standard accretion schemes which depend only on qc and qr. Although the authors present some interesting results with the scheme comparison, almost findings remained within the scope of the previous studies based on theoretical modeling (e.g., Wood, 2005; Wood et al., 2009; Lee and Baik, 2017) and climate

modeling (e.g., Gettelman et al., 2013, 2015). I feel that the authors should add more discussion about the physical mechanism behind the different scheme used.

Reply: Thanks for the suggestion. We have added more discussions with relevant studies, as listed below.
(1) Page 18, Line 356-358: The sign of the difference between the schemes is consistent with previous studies, e.g., Be68 and LD04 have larger autoconversion rate ($A_u$) than KK00 (Figures 7a and b) (Lee and Baik, 2017).
(2) Page 19, Line 382-385: Due to the weaker accretion in CP2k, fewer cloud droplets are collected by raindrops; these surviving cloud droplets are then available for autoconversion, which leads to the larger $A_u$ in CP2k (e.g. Gettelman et al., 2013; Posselt and Lohmann, 2008).
(3) Page 19-20, Line 387-398: Therefore, CP2k has the lowest ratio of accretion rate to autoconversion rate ($A_c/A_u$) with the mean value of 2.88 (2.81) over domain 02 (03) mainly because of small $A_c$. Bh94 has the largest ratio $A_c/A_u$ with a mean value of 151.24 (144.60) over domain 02 (03). Indeed, Bh94 exhibits the smallest $A_u$ (Figures 7e and f) of all schemes tested here. $A_c/A_u$ of all schemes is in the range of 0.1-296.3, consistent with previous studies (Gettelman et al., 2013; Lee and Baik, 2017; Michibata and Takemura, 2015; Seifert and Onishi, 2016; Jiang et al., 2010). In domain 03, the ratio of $A_c/A_u$ in CP2k correlates with    precipitation intensity (Figure 6b), consistent with the arguments about accretion-dominated and autoconversion-dominated regimes (Jiang et al., 2010; Wood et al., 2009; Michibata and Takemura, 2015). In domain 02, most of $A_c/A_u$ is larger than 1; some $A_c/A_u$ values are smaller than 1 but still with strong precipitation likely caused by the influence of ice/mixed-phase processes.
(4) Page 25, Line 498-503: Although $A_u$ in CP2k is larger than that in other schemes, $A_c$ ultimately determines the liquid-phase precipitation rate, which has been discussed in many previous studies (e.g., Jiang et al., 2010; Wood et al., 2009; Michibata and Takemura, 2015; Gettelman et al., 2015). The liquid-phase precipitation is suppressed by a weak $A_c$. Furthermore, large $A_u$ in CP2k can increase $q_r$ but decrease $q_c$, which may enhance or suppress $A_c$ (Posselt and Lohmann, 2008).
Besides, we also discuss the performance of the CP2k scheme and the physical mechanisms in reducing precipitation overprediction (Section 3.3 Page 19-25).

Therefore, I suggest major revisions and re-review. The authors also have presentation issues to address detailed below.

specific comments:

1. Lines 55-56 and 59-63: Is it true for the WRF model? Again, is this issue attributed

to the model resolution? Or microphysical parameterization?

Reply: Yes, it is true for the WRF model. This issue is attributed to both the model resolution and microphysical parameterization. For example, Xu et al. (2012) found that the WRF simulations could reproduce the trends of precipitation events but the intensities were doubled; they claimed that the low resolution was responsible for this phenomenon. Our results also indicate that resolution (Section 5) and microphysics (accretion scheme; Section 3.2-3.3) are two important factors affecting precipitation simulation over TP.

In order to highlight the main purpose of this study – liquid-phase processes, the paragraph starting with Line 59-63 in the original submission has been modified (Page 4, Line 55-66). We have added more references (Li et al., 2008; Gao et al., 2016; Chen et al., 2017a) to the sentence of Lines 53-56 in the original submission (Page 3, Line 51-53).

References

Sato, T., Yoshikane, T., Satoh, M., MIURA, H., and Fujinami, H.: Resolution dependency of the diurnal cycle of convective clouds over the Tibetan Plateau in a mesoscale model, Journal of the Meteorological Society of Japan. Ser. II, 86, 17-31, 2008.

Xu, J., Zhang, B., Wang, M., and Wang, H.: Diurnal variation of summer precipitation over the Tibetan Plateau: a cloud-resolving simulation, Annales Geophysicae, 2012, 1575-1586,

Maussion, F., Scherer, D., Finkelnburg, R., Richters, J., Yang, W., and Yao, T.: WRF simulation of a precipitation event over the Tibetan Plateau, China – an assessment using remote sensing and ground observations, Hydrology and Earth System Sciences, 15, 1795-1817, 10.5194/hess-15-1795-2011, 2011.

Gao, W., Sui, C. H., Fan, J., Hu, Z., and Zhong, L.: A study of cloud microphysics and precipitation over the Tibetan Plateau by radar observations and cloud‐resolving model simulations, Journal of Geophysical Research: Atmospheres, 121, 13,735-713,752, 2016.

Gerken, T., Babel, W., Sun, F., Herzog, M., Ma, Y., Foken, T., and Graf, H.-F.: Uncertainty in atmospheric profiles and its impact on modeled convection development at Nam Co Lake, Tibetan Plateau, Journal of Geophysical Research: Atmospheres, 118, 12,317-312,331, 10.1002/2013jd020647, 2013.

Li, Y., Wang, Y., Song, Y., Hu, L., Gao, S., and Rong, F.: Characteristics of Summer Convective Systems Initiated over the Tibetan Plateau. Part I: Origin, Track, Development, and Precipitation, Journal of Applied Meteorology and Climatology, 47, 2679-2695, 10.1175/2008jamc1695.1, 2008.

Chen, B., Hu, Z., Liu, L., and Zhang, G.: Raindrop Size Distribution Measurements at 4,500 m on the Tibetan Plateau During TIPEX‐III, Journal of Geophysical Research: Atmospheres, 122, 11,092-011,106, 2017a.

2. Line 82-84: This statement is too vague.

Reply: Sorry for the unclear statement. This sentence has been revised to make it clearer (Page 5 Line 75-80): "Gao et al. (2016) investigated the role of liquid-phase processes by excluding ice-phase microphysics, doubling the condensation rate, halving the evaporation rate and increasing the initial droplet radius, and found significant effects from all these sensitivity tests on the surface precipitation; they also suggested that liquid-phase rain processes could be more important than ice-phase processes over the precipitation cores during weak convection over the TP".

3. Line 86: Where did you state the entrainment-mixing issue? I did not find. This should be removed unless the authors provide data and discussion.

Reply: A sensitivity test is run assuming extremely inhomogeneous mixing mechanism (INHOMO in Table 3). Besides the INHOMO case, sensitivity tests of all autoconversion and accretion schemes in Table 3 are also conducted assuming homogeneous and extremely inhomogeneous mixing mechanisms, respectively, in the revised manuscript. Combined with the other referee comment, we have added discussions in the main text (Page 18, Line 364-369) and also in the supplement (Page 1-2, Line 9-22).
Please see the response to General Comment 1 for details.

4. Line 94-95: Please add reference(s) here.

Reply: We have added the following references: Grabowski (2006); Lu et al. (2013); Hoffmann and Feingold (2019) (Page 6, Line 107-108).

References
Grabowski, W. W.: Indirect impact of atmospheric aerosols in idealized simulations of convective–radiative quasi equilibrium, Journal of climate, 19, 4664-4682, 2006.
Lu, C., Liu, Y., Niu, S., Krueger, S., and Wagner, T.: Exploring parameterization for turbulent entrainment-mixing processes in clouds, Journal of Geophysical Research: Atmospheres, 118, 185-194, 10.1029/2012jd018464, 2013.
Hoffmann, F., and Feingold, G.: Entrainment and Mixing in Stratocumulus: Effects of a New Explicit Subgrid-Scale Scheme for Large-Eddy Simulations with Particle-Based Microphysics, Journal of the Atmospheric Sciences, 76, 1955-1973, 10.1175/jas-d-18-0318.1, 2019.

5. Line 120: Which satellite data did you use? Please add the description and appropriate citation(s).

Reply: Sorry for the unclear statement. We have revised the sentence and added some citations (Page 7, Line 124-131): "The simulations are compared against the precipitation dataset that Ma et al. (2018) derived from sparse gauge observations and multiple satellite precipitation datasets, including Tropical Rainfall Measuring Mission (TRMM) Multisatellite Precipitation Analysis (TMPA) 3B42RT and 3B42V7 (Huffman et al., 2007), Climate Prediction Center MORPHing technique (CMORPH) (Joyce et al., 2004) and Precipitation Estimation from Remotely Sensed Information using Artificial Neural Networks-Climate Data Record (PERSIANN-CDR) (Ashouri et al., 2015)".

References
Hoffmann, F., and Feingold, G.: Entrainment and Mixing in Stratocumulus: Effects of a New Explicit Subgrid-Scale Scheme for Large-Eddy Simulations with Particle-Based Microphysics, Journal of the Atmospheric Sciences, 76, 1955-1973, 10.1175/jas-d-18-0318.1, 2019.
Joyce, R. J., Janowiak, J. E., Arkin, P. A., and Xie, P.: CMORPH: A method that produces global precipitation estimates from passive microwave and infrared data at high spatial and temporal resolution, Journal of hydrometeorology, 5, 487-503, 2004.
Ashouri, H., Hsu, K.-L., Sorooshian, S., Braithwaite, D. K., Knapp, K. R., Cecil, L. D., Nelson, B. R., and Prat, O. P.: PERSIANN-CDR: Daily precipitation climate data record from multisatellite observations for hydrological and climate studies, Bulletin of the American Meteorological Society, 96, 69-83, 2015.

6. Line 148-149, Equations (1) and (2): The unit for mass tendency is kg/kg/s in general. Equation (5): Please define xc.

Reply: Thanks. We have changed the units, and added the definition of $x_c$: "$x_c$ is the normalized critical mass and can be written as a function of $N_c$ and $q_c$ (Liu et al., 2005)" (Page 10, Line 187-188).

7. Line 180: "most accretion schemes" needs a couple of references.

Reply: Taken. These references have been added:
Beheng, K.: A parameterization of warm cloud microphysical conversion processes,

Atmospheric Research, 33, 193-206, 1994.

Khairoutdinov, M., and Kogan, Y.: A new cloud physics parameterization in a large-eddy simulation model of marine stratocumulus, Monthly weather review, 128, 229-243, 2000.

Kogan, Y.: A cumulus cloud microphysics parameterization for cloud-resolving models, Journal of the Atmospheric Sciences, 70, 1423-1436, 2013.

8. Line 219-222: It is unclear because a lower precipitation rate is smoothed by blue. The color scheme in Figure 2 (and also Figure 5) should be modified. In the present figure, I cannot recognize the "rainband oriented in the northeast-southwest direction".

Reply: The color scheme has been changed to the rainbow scheme (both Figure 2 and Figure 5). We also label the maximum precipitation rate directly in Figure 2. The "northeast-southwest direction" is too subjective, therefore, we have deleted it.

9.Lines 347-349 and 353-355, and Figures 7b and 7d: I disagree with this sentence. The autoconversion rate in the CP2k scheme is higher (Figures 7a and 7c), whereas the accretion rate is the lowest. This is inconsistent with your explanation. And also, why the accretion rates are almost similar among various schemes except for the CP2k? If the same accretion scheme (KK00) is used except for the CP2k, it is natural that the accretion rates are almost the same among the other schemes. I am confused about that, and please describe a more detailed explanation.

Reply: After reading the references suggested by the referee (e.g., Wood, 2005; Wood et al., 2009; Gettelman et al., 2013, 2015), we realize that there is a complicated relationship between autoconversion rate and accretion rate. We have deleted the sentence of Lines 347-349 in the original submission. Instead, we add "Based on many previous studies (e.g., Seifert and Onishi, 2016; Lee and Baik, 2017; Gettelman et al., 2013; Wood et al., 2009; Jing et al., 2019), the relationship between $A_c$ and $A_u$ is nonmonotonic." (Page 19, Line 380-382)

The low accretion rate and high autoconversion rate in CP2k are explained (Page 19, Line 382-385): Due to the weaker accretion in CP2k, fewer cloud droplets are collected by raindrops; these surviving cloud droplets are then available for autoconversion, which leads to the larger $A_u$ in CP2k (e.g. Gettelman et al., 2013; Posselt and Lohmann, 2008).

For those with similar accretion rates in Figures 7c and 7d, two accretion schemes are used: the Ko13 scheme for the Ko13 sensitivity test, and the KK00 scheme for the

control run and the Be68, Bh94, LD04, INHOMO sensitivity tests. The reason why $A_c$ in the control run and other sensitivity tests is so comparable is that the two schemes have similar functions of rain and cloud water content (Eqs. 2 and 7) and similar variation trends in Figure 9. We have added these discussions in the revised manuscript (Page 19, Line 376-379)

10.Sections 3.2.2.1, 3.2.2.2, and 3.2.2.3: These sections should include more discussion. The current description only lists the result from different microphysical scheme use, but the variability due to the different treatment of autoconversion and accretion schemes has been well known (e.g., Wood, 2005; Wood et al., 2009; Gettelman et al., 2013, 2015; Jing et al., 2019). It would help readers if more in-depth discussions with relevant studies are shown in the text.

Reply: We agree with the referee that the variability due to the different treatment of autoconversion and accretion schemes has been well examined in the previous studies. More discussions are added in the revised manuscript and the references in the comment are all cited. Please see the response to General Comment 3.

In addition, the main finding of this study is that CP2k is unique and can better reduce precipitation overprediction. The results and physical mechanisms are discussed in Sections 3.3 and 4 (Page 19-26).

11.Line 367-368: The difference of Nc between the INHOMO run and CTRL run is quite small. It is better to conclude that the impact of the entrainment-mixing is less important than the liquid conversion process in this study, although it depends strongly on the cloud type simulated in the TP area. Please add more discussion here (or conclusion section).

Reply: Thank you for the suggestion. We have added discussions in the main text (Page 18, Line 364-369) and also in the supplement (Page 1-2, Line 9-22). Please see the response to General Comment 1 as well for more.

12.Line 400-404: This may imply that the uncertainties in ice- and mixed-phase microphysics schemes are conveyed to the uncertainty of surface precipitation. To confirm this, an additional simulation without the riming process will be helpful to understand the microphysical mechanisms.

Reply: Thanks for the great suggestion. We have carried out the CP2k sensitivity test without the riming process. Figure R1 shows the microphysical processes conversion rates in CP2k with riming minus those without riming. Riming suppresses the liquid-phase rain formation processes through reducing $A_c$, but enhances ice/mixed-phase rain formation processes through increasing melting rate. The sensitivity of warm/cold rain formation to riming ultimately trickles down to uncertainties in the simulation of surface precipitation. We have added some discussions in the main text (Page 20, Line 407-412) and Figure R1 is added in the supplement.

[Figure]

Figure R1. Differences of mean vertical profiles of the dominated microphysical processes conversion rates between the case with riming process and the case without riming process in CP2k (CP2k with riming minus CP2k without riming) from (a) domain 02 except southeastern corner, (b) the southeastern corner of domain 02, and during the two precipitation peak periods (c) 0700-1200 UTC 22 July and (d) 0700-1200 UTC 23 July over domain 03. The purple dot-dash lines denote the mean height of 0 ℃ isotherm.

13.Line 424-426: Does it mean that the other microphysical schemes except for the CP2k overestimate the accretion rate and consequent precipitation? Or is this the case only for the TP region? I suggest that the authors refer to the ratio of accretion over autoconversion (e.g., Gettelman et al., 2013; Seifert and Onishi, 2016; Lee and Baik, 2017) among the experiments. This metric will be helpful to evaluate the dependence of microphysical process rates on cloud regimes in the model.

Reply: Yes, the other microphysical schemes except for the CP2k overestimate the accretion rate and consequent precipitation, based on the TP case from 22 to 23 July, 2014 and the one-month simulations from 22 July to 21 August, 2014 (Section 4, Page 25-26). We would argue that the conclusion is valid for precipitation in other regions beyond TP, according to the theoretical analysis in Section 3.3.2. We add these discussions in the revised manuscript (Page 30, Line 613-616). Especially, the sentence "More studies are needed to understand whether these findings are applicable to regions beyond the Tibetan Plateau as well" is added.

Thanks for the suggestion of examining the ratio of accretion rate to autoconversion rate. We have examined it and discussed the results in revision (Page 19, Line 387-398): CP2k has the lowest ratio $A_c/A_u$ with the mean value of 2.88 (2.81) over domain 02 (03) mainly because of small $A_c$. Bh94 has the largest ratio $A_c/A_u$ with a mean value of 151.24 (144.60) over domain 02 (03). Indeed, Bh94 exhibits the smallest $A_u$ (Figures 7e and f) of all schemes tested here. $A_c/A_u$ of all schemes is in the range of 0.1-296.3, consistent with previous studies (Gettelman et al., 2013; Lee and Baik, 2017; Michibata and Takemura, 2015; Seifert and Onishi, 2016; Jiang et al., 2010). In domain 03, the ratio of $A_c/A_u$ in CP2k correlates with precipitation intensity (Figure 6b), consistent with the arguments about accretion-dominated and autoconversion-dominated regimes (Jiang et al., 2010; Wood et al., 2009; Michibata and Takemura, 2015). In domain 02, most of $A_c/A_u$ is larger than 1; some $A_c/A_u$ values are smaller than 1 but still with strong precipitation likely caused by the influence of ice/mixed-phase processes.

14.Lines 483-485 and 491-492: Why does the CP2k scheme improve the too early onset of precipitation in spite of the higher autoconversion rate compared with the other schemes?

Reply: There are two reasons. First, CP2k has a more physically reasonable threshold of cloud effective radius to initiate precipitation. In contrast, the control run and Ko13 initiate precipitation too early. The liquid-phase precipitation rate exceeds 2 mm/day when the cloud effective radius is 9 μm in the control run and Ko13. In CP2k, it is not until the cloud effective radius reaches about 15 μm, that the precipitation rate exceeds 2 mm/day. The value of 9 μm, is much smaller than 14 μm that is needed to initiate liquid-phase precipitation, often suggested by observational studies (Rosenfeld et al., 2019).

Second, Although $A_u$ in CP2k is larger than that in other schemes, $A_c$ ultimately determines the liquid-phase precipitation rate, which has been discussed in many previous studies (e.g., Jiang et al., 2010; Wood et al., 2009; Michibata and Takemura, 2015; Gettelman et al., 2015). The liquid-phase precipitation is suppressed by a weak $A_c$. Furthermore, large $A_u$ in CP2k can increase $q_r$ but decrease $q_c$, which may enhance or suppress $A_c$ (Posselt and Lohmann, 2008). In other schemes, the accretion process is triggered to a considerable amount with small liquid drops due to the overestimation of $A_c$ when confined to small drops (Page 25, Line 498-505). As shown in Figure 9, the $A_c$ in the KK00 or the Ko13 scheme is always larger than that in CP2k when the raindrop radius is smaller than 2000 μm. The difference between CP2k and the other two schemes increases with decreasing raindrop radius; especially when the raindrop radius is smaller than 50 μm, with the maximum difference being more than two orders of magnitude. Figure 10 shows the PDFs of raindrop radius used in the accretion process in the three schemes. All raindrops are smaller than $10^3$ μm. The PDFs have peaks of ~30, ~30, and ~25 μm in the control run, Ko13, and CP2k, respectively, and the cumulative PDF shows that the raindrops with radius smaller than 50 μm have frequencies of 58.8%, 53.8%, and 46.0%, respectively. The drop size distributions from both aircraft observations and bin models also confirm that a large proportion of liquid droplets have radii larger than 25 μm but smaller than 50 μm (Wood, 2005b; Morrison and Grabowski, 2007). Such a large percentage of small raindrops makes the $A_c$ and precipitation in CP2k quite different from that in other schemes (Figure 9). Furthermore, there is a positive feedback mechanism, since accretion increases $q_r$ and $A_c$ is positively correlated with $q_r$. The overestimation of the $A_c$ in KK00 or Ko13 hence feeds back on itself.

The above discussions are shown in Section 3.3.2. (Page 22-25)

Figure 1: Please explain the color shade.

Reply: The explanation has been added: "The color bar represents the height (m) above the sea level."

Figures 2 and 5: Unclear color contrast. Please consider to change the current color scheme (blue-to-red) to standard rainbow color or white-to-blue etc.

Reply: Thank you for your suggestion. The color scheme has been changed to rainbow scheme.

---

## Author Comment (AC2) · 25 Mar 2020

**Response to Anonymous Referee #2**

**1.** I have reviewed "Effects of Liquid Phase Cloud Microphysical Processes in Mixed Phase Cumulus Clouds over the Tibetan Plateau" by Xu et al. I do not have any complaints about the analysis itself, but I have serious doubts that the manuscript is sufficiently relevant beyond the one case investigated to be within the scope of ACP. The article describes an analysis performed on a single synoptic system transiting the Tibetan Plateau. The authors focus on warm cloud processes, performing sensitivity studies with different autoconversion/accretion/droplet evaporation parameterizations; why they focus on these processes is not well explained, in particular since the precipitation in their study case is clearly initiated in the ice phase (figure 4), so one would expect that only accretion and ice/mixed-phase processes matter.

Reply: Thanks for the positive evaluation, and we respectfully disagree on the relevance comment. Yes, accretion and ice/mixed-phase processes matter. However, autoconversion and entrainment-mixing processes could be important as well, esp. in context of understanding the outstanding problem of overprediction of precipitation over the Tibetan Plateau. The reason is that accretion is the collection process between rain drops and cloud droplets, which are significantly affected by rain/cloud liquid water content and number concentration. These microphysical properties are influenced by autoconversion and entrainment-mixing processes. These three processes are likely to be intertwined.

So far, it is still unknown how the three liquid-phase processes affect TP precipitation, and whether improving the parameterizations of these three liquid-phase processes can mitigate the problem of overpredicted precipitation. Also unknown is which liquid-phase process is the most important in affecting TP precipitation, and which commonly used scheme can best describe the most important process and why. To address the concern, we have reframed the structure of introduction (Page 4, Line 55-66; Page 5, Line 81-90).

In addition, to further highlight the main purpose of this study, the phase "precipitation overprediction" is emphasized in all sections in the revised manuscript. For example, the titles of Sections 3.2 and 3.3 are modified to be "Sensitivity of precipitation overprediction to different liquid-phase processes" and "Reasons for improvements of precipitation overprediction in CP2k", respectively.

According to the referee comment, we have significantly shortened the description of the simulations with different autoconversion and entrainment-mixing schemes. Please see the response to Comment 2.

**2.** Not surprisingly, the authors find that autoconversion and homogeneous vs inhomogeneous cloud droplet evaporation make very little difference in accumulated precipitation. In a revised manuscript, I would suggest getting rid of several pages of unsurprising results and replacing them simply with one paragraph along the lines of, "we analyzed the effect of different autoconversion parameterizations and mixing assumptions and found them to have no substantial impact."

Reply: Thank you for your suggestions. We have significantly shortened the detailed description from several pages to less than one page (Line 349-369). By combining Comment 3 and the other referee comment, we also add some discussion with previous studies in a very concise way (Supplement, Page 1-2, Line 9-22).

**3.** The finding that accretion is an important control on accumulated precipitation is also not very surprising. Furthermore, it is not clear what we are supposed to do with this information. When parameterizations are developed, they are usually tuned to do something reasonable in one or a handful of test cases, but it is understood that they will probably not give results that match observations in every conceivable case – usually far from it. So it is not surprising that some parameterizations do better than others at reproducing this particular case. However, that does not mean that the winner in this case will produce the best results in other cases. Are the authors recommending that the Cohard and Pinty (2000) accretion parameterization should be used generally, or generally for Tibetan Plateau studies? How does one case study support that recommendation? If that is not the recommendation, what is new or useful about the results? That different warm cloud microphysics schemes can lead to wildly different simulations of individual cases is nothing new; for an example of a study that draws this conclusion in a more generalized way, with interesting statements about science implications, see White et al. (2017), https://doi.org/10.5194/acp-17-12145-2017
Thus, I recommend that the authors substantially revise their manuscript to focus on conclusions that are of use beyond this one case study. If this is not possible, I do not think that ACP is the appropriate journal.

Reply: Thank you for the suggestion. We have revised the paper accordingly. Briefly, month-long simulations and related discussions are added in Section 4 in the revised manuscript (Page 25-26). Figure R1 and Table R1 are also added in the revised manuscript. We design one-month simulations which start from 0000 UTC 21 July to 0000 UTC 21 August, 2014 with the same domains using the three accretion schemes (the KK00 scheme in the control run, CP2k, and Ko13) to figure out whether the mitigation of overprediction on precipitation from the CP2k only works in the particular

case in Section 3 or is generally valid.

Figure R1 shows the temporal evolution of the area-averaged daily precipitation rate in domains 02 and 03 from the three accretion simulations and the observations. Compared with the observed precipitation, the control run significantly overestimates precipitation for most days, especially in domain 02. The results of Ko13 are very close to those in the control run, while CP2k significantly reduces precipitation overprediction with $p$-values of student's t-test less than 0.01 for both domain 02 and domain 03. The average precipitation rate in the observation, the control run, Ko13, and CP2k are, respectively, 1.56, 2.46, 2.49, and 2.17 mm/day over domain 02, and 4.54, 5.80, 5.87, and 5.17 mm/day over domain 03. These numbers confirm the better performance of CP2k than the other accretion schemes. Table R1 shows that CP2k has higher HSS scores than the control run and Ko13 over both domains 02 and 03.

Therefore, the effects of CP2k on reducing precipitation overprediction are not limited to our specific case. It is recommended that the CP2k accretion parameterization should be used generally, at least for Tibetan Plateau studies. We would argue that the conclusion is valid in other regions beyond the Tibetan Plateau, according to the theoretical analysis in Section 3.2.2. We add these discussions in the revised manuscript (Page 30, Line 610-616). Especially, the sentence "More studies are needed to understand whether these findings are applicable to regions beyond the Tibetan Plateau as well " is added.

We also agree that many studies have found that different warm cloud microphysics schemes can lead to different simulations of individual cases. Combined with the other referee comment, we have added some discussions and White et al. (2017) is cited (Page 25, Line 508-511; Page 6, Line 101).

[Figure]

Figure R1. Time series of area-averaged daily precipitation rate (mm/day) from 0000 UTC 22 July to 0000 UTC 20 August 2014 over (a) domain 02 and (b) domain 03 in the observations and three accretion cases (the control run, CP2k and Ko13).

Table R1. The values of four elements a-d and Heidke skill score (HSS) for three one-month simulations over domain 02 and domain 03 of the control run and CP2k, Ko13 (different accretion schemes).

|  | $a$ | $b$ | $c$ | $d$ | HSS |
|---|---|---|---|---|---|
| domain 02 |  |  |  |  |  |
| Control | 3780 | 5052 | 1924 | 24584 | 0.403 |
| CP2k | 3749 | 4369 | 1955 | 25267 | 0.435 |
| Ko13 | 3764 | 4825 | 1940 | 24811 | 0.413 |
|  |  |  |  |  |  |
| domain 03 |  |  |  |  |  |
| Control | 1188 | 2856 | 93 | 2538 | 0.220 |
| CP2k | 1163 | 2355 | 118 | 3084 | 0.262 |
| Ko13 | 1181 | 2908 | 100 | 2531 | 0.211 |